# Chromosome compartment assembly is essential for subtelomeric gene silencing in trypanosomes

Luiza Berenguer Antunes[1,3], Tony Isebe [1,3], Oksana Kutova[1] & Igor Cestari [1,2] ✉

Genome three-dimensional organization is essential for eukaryotic gene expression. The chromosomes of the pathogen *Trypanosoma brucei* contain hundreds of silent variant surface glycoprotein (VSG) genes in subtelomeric regions. *T. brucei* transcribes a single VSG gene and periodically changes the VSG expressed, altering its surface coat to escape host antibodies by antigenic variation. We show that *T. brucei* core and subtelomeric chromosome compartments are separated by distinct boundaries and display topologically associating domains and loops. Chromosomes co-interact through compartment boundaries, which insulate silent subtelomeric from transcribed core compartments. We uncover chromatin-associating factors at the boundaries, including repressor-activator protein 1 (RAP1), which spreads over silent compartments. Inactivation of the RAP1 regulator, phosphatidylinositol phosphate 5-phosphatase, removes RAP1 from boundaries and subtelomeric compartments, disrupting chromatin compartment contacts and activating all VSG genes. The data show spatial segregation of repressed from transcribed chromatin and phosphoinositide regulation of compartment assembly and genome organization.

Eukaryote chromosomes are typically organized into megabase (Mb) scale compartments defined as transcribed A compartment (euchromatin) and silent B compartment (heterochromatin)[1,2]. Compartments are further organized into topologically associating domains (TADs) and loops[2,3]. This spatial organization provides levels of compaction and coordination of gene expression by placing spatially distal genomic regions in proximity, such as genes and their promoters and enhancers, or spatially parting transcribed from repressed regions[4–6]. The Mb-long diploid chromosomes of the protozoan pathogen *Trypanosoma brucei* are organized into core regions containing polycistronic units (PTUs) encoding housekeeping and hypothetical proteins arranged as directional gene clusters, whereas subtelomeric regions contain largely silent variant surface glycoprotein (VSGs) genes, and, in some chromosomes, telomere-proximal expression sites (ESs) from where VSG genes can be transcribed[7]. The PTUs contain dozens of genes lacking canonical RNA polymerase II promoters and are co-transcribed into polycistronic RNAs processed by trans-splicing[8].

*T. brucei* infects humans and animals, causing sleeping sickness. These parasites express a homogeneous surface coat composed of ~$10^7$ VSG proteins[9,10], which they periodically switch to evade host antibody clearance by antigenic variation. VSG proteins contain a variable surface-exposed N-terminal domain, and a conserved C-terminal region attached to the cell's surface by a glycosylphosphatidylinositol anchor[11,12]. There are over 2500 VSG genes and pseudogenes in this organism's genome, found primarily in subtelomeric regions, which function as a storage site for VSG sequences[7]. However, a single VSG gene is transcribed at a time from one of the 20 ESs. The change in the VSG expressed occurs by transcriptional switching among ESs or by recombination of VSG genes within the

[1]Institute of Parasitology, McGill University, Sainte-Anne-de-Bellevue, Quebec, QC, Canada. [2]Division of Clinical and Translational Research, Department of Medicine, McGill University, Montreal, QC, Canada. [3]These authors contributed equally: Luiza Berenguer Antunes, Tony Isebe. ✉e-mail: igor.cestari@mcgill.ca

active ES, altering the parasite surface coat. *T. brucei* chronic infection relies on the vast repertoire of subtelomeric VSG genes to escape host antibodies and VSG monogenic expression. The parasites' inability to maintain their repression results in multiple VSGs expressed[13,14] and their immune clearance during infection[15]. The mechanisms regulating VSG monogenic expression are incompletely understood. The active VSG is transcribed from a nuclear site termed ES body (ESB)[16]. Specific factors, such as ESB-1 and VSG exclusion 1 and 2, associate with the active ES in the ESB and are required for VSG gene transcription[17–19]. In contrast, the silencing of the remaining ESs depends on repressor-activator protein 1 (RAP1)[20], which binds to 70 bp and telomeric repeat sequences flanking the ES VSG genes[13,14]. RAP1 silencing function is controlled by a phosphoinositide regulatory system, disruption of which results in VSG derepression and switching[14,21]. Specifically, PI(3,4,5)P3 is an allosteric regulator of RAP1, and its nuclear accumulation displaces RAP1 from ESs, resulting in VSG transcription[13,14]. A nuclear phosphatidylinositol phosphate 5-phosphatase (PIP5Pase) dephosphorylates PI(3,4,5)P3 and maintains ES repression. Its catalytic inactivation results in ES transcription and VSG switching[14], indicating that PIP5Pase plays a role in ES chromatin regulation.

Chromatin conformation capture experiments showed that core and subtelomeric regions of *T. brucei* chromosomes form distinct compartments[7], which correlate with their transcribed and repressed state, respectively. In the core regions, highly transcribed chromatin, such as that encoding tubulin and the transcribed ES, is proximal to the splice-leader array[22,23]. This arrangement might facilitate mRNA processing by trans-splicing[22]. Moreover, chromosomes were shown to be arranged around polymerase-specific transcription hubs, such as clusters formed by RNA polymerase II transcription start regions (TSRs) proposed to be formed by a process distinct from cohesin-mediated loop extrusion[24]. Although this chromosome organization is reminiscent of other eukaryotic chromatin folding[3,25,26], it might have evolved to accommodate polycistronic transcription. While these studies provide insights into the organization of transcribed regions of *T. brucei* chromosomes, much less is known about the organization of silent subtelomeric regions. These regions are significant because they harbour the silent VSG repertoire used for antigenic variation. Specifically, it remains unknown what delimits the VSG-rich subtelomeric silent compartments from core transcribed regions and how the subtelomeric compartments are repressed.

We show that silencing subtelomeric VSG genes in *T. brucei* entails chromatin spatial organization. Using cross-linking mass spectrometry (XLMS) and ChIP-seq, we identified signalling and chromatin-regulatory proteins associated with the compartment boundaries that separate silent from transcribed regions, including RAP1, histone deacetylase 1 (HDAC1), histone acetyltransferase (HAT1), and bromo-domain factor 2 (BDF2), among others. Moreover, we show that compartment boundaries of various chromosomes co-interact, likely helping to insulate silent from active chromatin regions. In addition to the boundaries, we show that RAP1 spreads over silent subtelomeric compartments in all chromosomes. The knockdown of the RAP1 regulator, PIP5Pase, disrupts chromatin interactions within compartments and displaces RAP1 from compartment boundaries and subtelomeric regions, resulting in the transcription of all subtelomeric VSG genes. The data show factors associated with chromosome compartment boundaries that segregate core from subtelomeric regions and indicate that silencing VSG-rich subtelomeric regions is RAP1-dependent. It also shows a role for phosphoinositides in controlling chromatin spatial organization.

## Results

### Signalling and chromatin regulatory cross-link network at ~ 30 Å resolution

We performed in vivo chemical cross-linking and immunoprecipitation of PIP5Pase followed by mass spectrometry (XLMS) to identify additional proteins potentially involved in signalling and chromatin regulation. We used a *T. brucei* bloodstream form line expressing an endogenously tagged PIP5Pase with three C-terminal V5 epitopes (42 amino acids), which we showed do not interfere with enzyme activity, interactions, or localization[13,14,21]. We cross-linked cells with disuccinimidyl suberate (DSS), a cell-permeable non-cleavable cross-linker with amine-reactive N-hydroxysuccinimide (NHS) esters flanking an 11.4 Å spacer arm (Fig. 1A). The NHS-esters react with primary amines of amino acids, typically lysines, but can also react with serines, tyrosines, and threonines[27] cross-linking amino acids within a 20–30 Å proximity[28]. DSS (0.25 mM) treatment of cells resulted in protein cross-linking, as indicated by a characteristic molecular weight (MW) shift observed by SDS/PAGE and Western blotting (Fig. 1B, C). Immuno-precipitation with anti-V5 antibodies resulted in the enrichment of PIP5Pase-V5 (Fig. 1C). MS analysis from 11 experiments identified 158,718 total peptides, of which 41.4% were inter-links among different proteins, 0.7% were intra-links (mono or loop links), and 57.9% were not cross-linked (Fig. 1D). A total of 50,157 cross-linked protein pairs were identified with 1% false discovery rate (FDR). A mean reproducibility index of $6.11 \pm 3.59$ was detected for total cross-linked proteins and $6.29 \pm 2.79$ for nuclear proteins, i.e., on average, a cross-linked protein was reproduced in 6 experiments with a mean detection frequency of $12.41 \pm 14.68$ and a mean of $13.2 \pm 14.1$ cross-links per protein (Supplementary Fig. 1 and Supplementary Data 1). Identifying cross-linked peptides can be challenging, requiring efficient in vivo protein cross-linking, peptide digestion, and mass spectrometry detection[29]. The high number of cross-linked and non-cross-linked peptides obtained indicates sufficient DSS cross-linking, immunoprecipitation, and protein digestion for mass spectrometry detection. Enrichment analysis comparing the immunoprecipitations in cells expressing V5-tagged PIP5Pase or non-tagged PIP5Pase (as a control) confirmed the enrichment of PIP5Pase (log2 fold-change = 3.66, *p* value = 0.00029) (Supplementary Fig. 2 and Supplementary Data 2). To validate the cross-link dataset, we analyzed whether the cross-linked proteins overlapped with PIP5Pase co-immunoprecipitated proteins that we previously identified without cross-linking[13]. The data show inter-protein cross-links for the top 30 proteins previously identified in PIP5Pase immunoprecipitations without cross-linking. The overlapping dataset confirms PIP5Pase enrichment and suggests potential direct interactions between proteins (Supplementary Fig. 2). The larger number of proteins identified by XLMS compared to those identified by standard (not cross-linked) immunoprecipitations[13] may result from cross-links between proximal or transiently interacting proteins not detected by standard approaches.

The data revealed a nuclear cross-linked network of 494 proteins with 1713 cross-links and 632 cross-link pairs involved in nucleic acid processes (Supplementary Data 3), including chromatin regulation, kinetochore, transcription and splicing, nuclear pore, and signaling and regulatory proteins such as protein kinases and phosphatases (Fig. 1E). Analysis of bona fide nuclear proteins revealed ~80% cross-linked with other nuclear proteins, indicating that most organellar cross-links occurred primarily before cell lysis (Supplementary Fig. 1). Analysis of PIP5Pase direct and counterpart cross-links revealed a subnetwork of 204 proteins with 750 cross-links (Fig. 1F and Supplementary Data 4). The large number of cross-linked proteins might reflect the proximity of proteins, captured as a chain of cross-links among various proteins, but it does not imply direct PIP5Pase protein interactions. There were 25 proteins with multiple cross-links, suggesting that these proteins may have different interactions, perhaps as part of protein complexes or in various subnuclear locations, concurring with PIP5Pase localization (Fig. 1F and Supplementary Data 4). Notably, the network was enriched in proteins involved in chromatin regulation, including histone acetylases (HAT) and deacetylases (HDAC), imitation switch (ISWI), bromodomain factors (BDF), histone methyltransferases (DOT1B), and RAP1

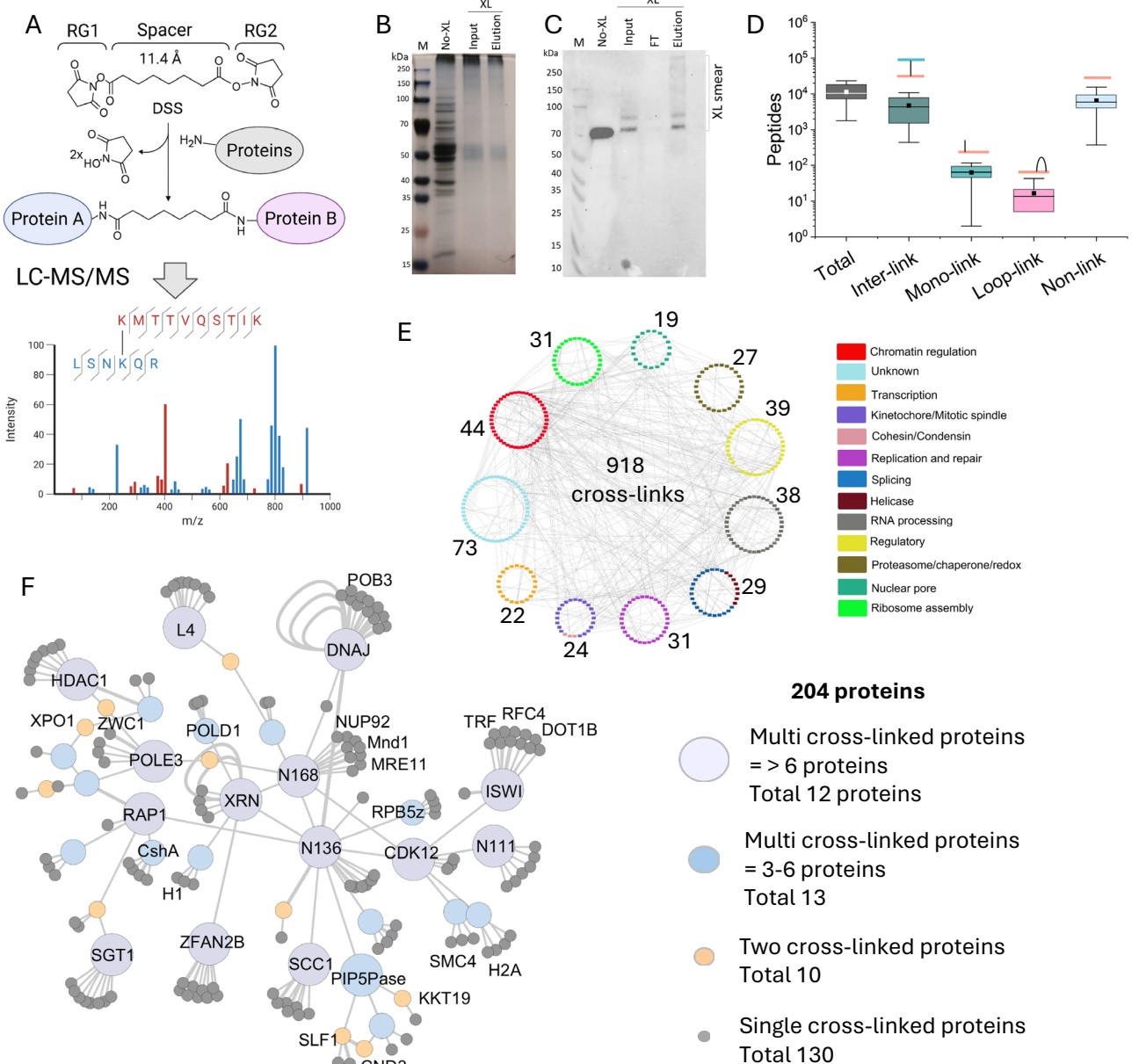

**Fig. 1 | PIP5Pase-V5 protein interaction network by XLMS. A** Diagram of disuccinimidyl suberate (DSS) cross-linker, its reaction with proteins, and cross-linked peptide detection by Liquid Chromatography-Tandem Mass Spectrometry (LC-MS/MS). RG1 and RG2, NHS-reactive groups 1 and 2. Created in BioRender. Cestari, I. (2025) https://BioRender.com/7g1lv3n. **B-C**) Silver-stained 10% SDS/PAGE **B** and Western blot **C** of immunoprecipitated V5-tagged PIP5Pase with α-V5 monoclonal antibodies (mAb) from a lysate of DSS cross-linked (XL) *T. brucei* bloodstream forms. Western blot probed with α-V5 mAbs. M protein marker, FT flow-through. **D** Number and types of cross-linked peptides detected by mass spectrometry. The box represents the interquartile range (IQR, 25–75%); the whiskers with minima and maxima show the range of smallest to largest values; the center line is the median, and the square symbol is the mean. **E** Cross-links of a subset of nuclear proteins (377 proteins and 918 cross-links) identified by PIP5Pase immunoprecipitation, organized by functional categories. The number of proteins in each category is indicated. **F** PIP5Pase cross-link interaction network. Edges represent cross-links, and nodes represent proteins. See Supplementary Data 3 for abbreviations and protein description. **D**–**F** show the combined XLMS data of 11 biological replicates with an FDR of 1%. Images in **B** and **C** are representatives of 11 experiments. Source data are provided as a Source data file.

(Fig. 1E, F and Supplementary Data 4). It also included DNA replication and repair proteins, such as mini-chromosome maintenance (MCM) complex and replication factor C (RepC) subunits; several transcription and RNA processing factors, including class I transcription factor A (CITFA) complex subunits, and signalling/regulatory proteins, such as cdc2-related kinase 9 (CRK9) and protein phosphatase 1 (PP1). The cross-links suggest multiple processes concurring with PIP5Pase's subnuclear location. Many of these proteins have been associated with VSG expression and switching[8], consistent with PIP5Pase's role in VSG gene regulation[13,14,21]. The

PIP5Pase cross-linking network may reflect transient and stable interactions among proteins in various chromatin-regulatory processes[13,14,21], perhaps resulting from molecular crowding and dynamic protein associations with the chromatin.

### Mapping protein-interacting domains with XLMS
Analysis of cross-linked peptides identified potential interacting domains among several proteins involved in signalling and chromatin-regulatory processes (Fig. 2A–C). PIP5Pase cross-links were surface exposed and mainly distributed through its N-terminus (aa 1–150),

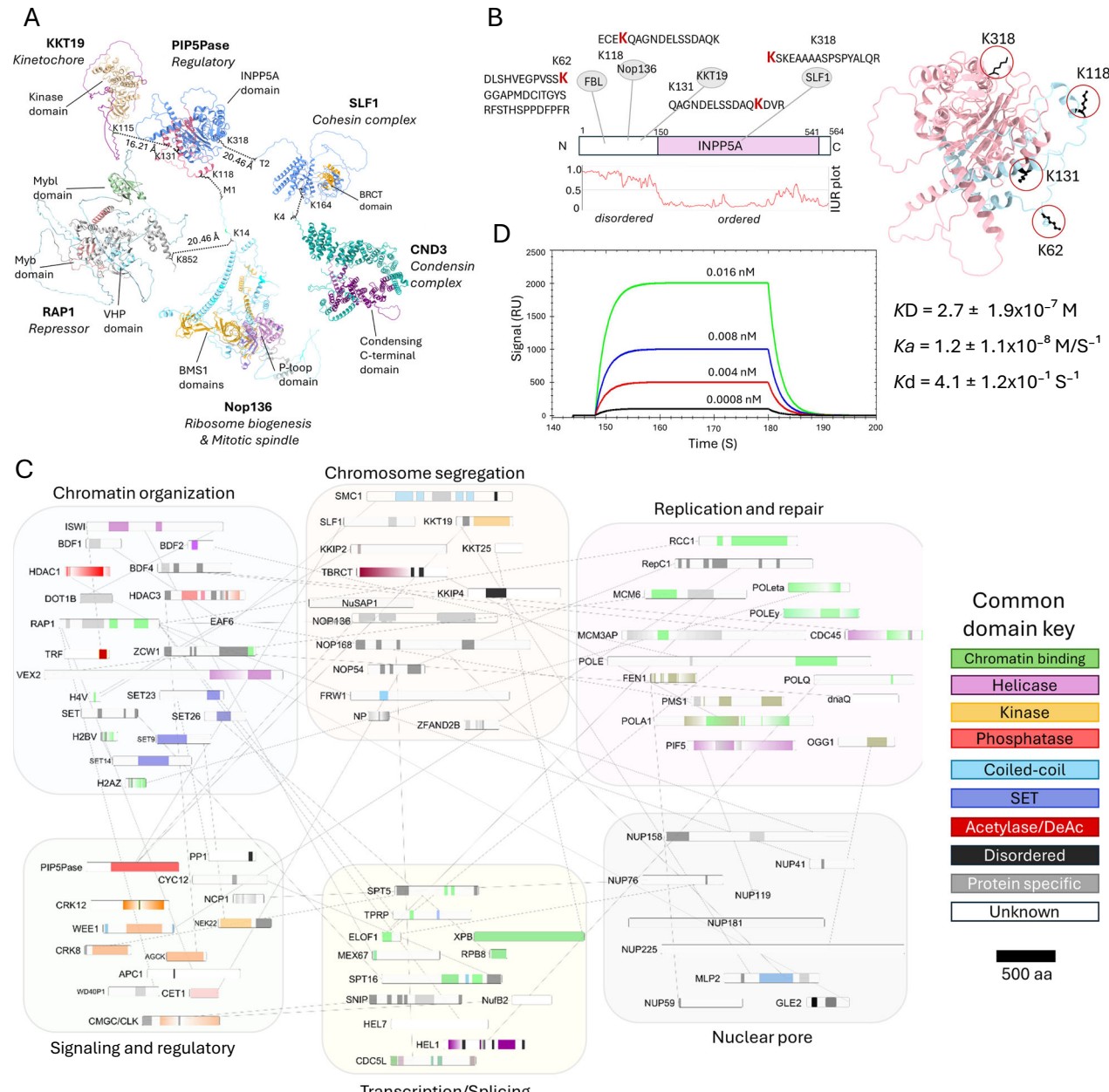

**Fig. 2 | Mapping potential protein-interacting domains with XLMS. A** Selected DSS cross-links between PIP5Pase (AlphaFold ID AF-Q385E2-F1-v4), KKT19 (AF-Q382T9-F1-v4), SLF1 (AF-Q57YW5-F1-v4), CND3 (AF-Q57ZK1-F1-v4), RAP1 (AF-Q387L4-F1-v4), and Nop136 (AF-Q384C0-F1-v4) are indicated in the AlphaFold predicted protein structures. Dotted lines indicate cross-linked residues. The cross-link distance between the two proteins was obtained using ChimeraX analysis of AlphaFold and cross-linked peptides[72]. Protein domains are indicated in colour. Protein function is indicated. **B** PIP5Pase diagram and cross-linked residues. The AlphaFold predicted structure (AF-Q385E2-F1-v4) is shown on the right with cross-linked residues indicated. In pink, INPP5A domain; in blue, N-terminal region. **C** PIP5Pase cross-link network of selected chromatin-associated proteins showing cross-linked protein regions. Protein domain function is indicated by colour. See Supplementary Data 3 for gene ID, product name abbreviations, and cross-linked residues. XLMS data is from 11 biological replicates with an FDR of 1%. **D** Native PIP5Pase-V5 and recombinant RAP1-His interactions by SPR. RAP1-His was bound to sensor, and PIP5Pase-V5 (0.0008, 0.004, 0.008, 0.016 nM) was added. Data from three biological replicates. $K_D$, equilibrium dissociation constant; $K_d$, dissociation rate constant; $K_a$, association rate constant. Data show mean ± standard deviation of the mean (SDM). IDR, intrinsically disordered regions. INPP5A inositol polyphosphate-5-phosphatase A domain, BMS1 BMS1 Ribosome Biogenesis Factor, VHP villin headpiece, BRCT BRCA1 C-Terminal domain. Myb Myb DNA-binding domain.

which is consistent with the N-terminus containing a predicted disordered region and the C-terminus containing the 5-phosphatase catalytic domain (Fig. 2B). It cross-linked with the N-terminus of KKT19, a kinetochore complex Cdc2-like protein kinase[30], and with the cohesin subunit protein SMC5-SMC6 complex localization factor protein 1 (SLF1), which functions in chromatin organization and DNA break and repair[2,31,32] (Fig. 2A, B and Supplementary Data 4). SLF1 cross-linked with condensin subunit 3 (CND3) and a putative BRCA2 and CDKN1A-

interacting protein (BCIP1) (Fig. 2B), which also functions in chromosome segregation. The data revealed cross-links in multiple chromatin-associated factors (Fig. 2C), revealing the proximity of proteins involved in chromatin organization, transcription, and splicing, chromatin regulatory proteins, DNA replication, and repair (Fig. 2C). The identified cross-linked amino acids show the potential interacting domains among these proteins (Fig. 2C) and how the network might co-interact. Many cross-linked proteins were previously shown to co-

interact[13,33], such as PIP5Pase, RAP1, HDAC3, BDF2, ZCW1, EAF6, TRF, DNA PolQ, SMC1, TRF, H2A, NUP158, and GLE2. Their reproducibility here validates the cross-link dataset; however, the cross-link data indicate potential direct partners and their protein domains involved in interactions, thus providing a more refined dataset.

Our previous work suggested that PIP5Pase interacts with RAP1[13,21]. Immunoprecipitations of V5-tagged PIP5Pase co-immunoprecipitated RAP1 (Supplementary Fig. 2 and Supplementary Data 2), confirming earlier observations[13,21,33]. We obtained 373 pep-tides for PIP5Pase, of which only six were cross-linked. In contrast, we obtained 241 peptides for RAP1, of which 39 were cross-linked, indi-cating that PIP5Pase is a particularly poorly cross-linked protein by DSS. In vitro cross-linking after PIP5Pase immunoprecipitation identi-fied 57 peptides for PIP5Pase but did not result in additional cross-links (Supplementary Data 5). Because we obtained 158,718 total peptides, of which 66,820 were cross-linked, it is unlikely that this reflects inef-ficient digestion of cross-linked proteins, but perhaps an intrinsic limitation of this protein in being cross-linked by DSS. The limitations could be due to its folding, thereby decreasing the exposure of DSS cross-linkable amino acids. Nevertheless, its enrichment revealed a chain of cross-links, indicating a protein interaction network (Figs. 1F and 2C). Analysis of RAP1 cross-links showed a distribution to its N- and C-terminus (from amino acids 2 -13 or 847- 852) and distal from the villin headpiece (VHP) or the Myb and Myb-like domains (Fig. 2C and Supplementary Data 4), which we showed bind to PI(3,4,5) P3 and DNA, respectively[14]. The protein Nop136, which has multiple disordered regions and functions in nucleolus assembly and chromo-some segregation[34], cross-linked with PIP5Pase and RAP1 (Fig. 2A–C), suggesting spatial proximity among them. Binding kinetics by surface plasmon resonance (SPR) using *T. brucei* his-tagged RAP1 recombi-nantly produced in *E. coli* and V5-tagged PIP5Pase isolated from *T. brucei*[13,14,21] showed that PIP5Pase and RAP1 interacted directly with a $K_D$ of 2.7 nM (±1.9 nM), confirming their proximity by cross-link and showing that these proteins can interact directly. The absence of cross-link between the two proteins may result from the stringency of the in vivo cross-link approach, i.e., two cross-linkable amino acids (aa) within ~20−30 Å proximity. The XLMS data revealed a network of sig-nalling and chromatin-regulatory proteins and their potential inter-acting domains in vivo. The cross-linking among proteins involved in transcription, splicing, chromatin organization, nuclear pore, DNA replication, and repair indicates some of these processes might concur (at a given time) at specific chromatin regions where PIP5Pase is found. The data suggest that PIP5Pase might have a broader role in chromatin regulation in addition to its function controlling telomeric ES silencing[13,14,21].

**Interacting proteins at chromosome compartment boundaries**

We postulated that the cross-links between chromatin-regulatory proteins reflect their spatial proximity due to chromatin's three-dimensional organization. Thus, we performed Hi-C and ChIP-seq analysis in *T. brucei* bloodstream forms. *T. brucei* chromosomes are organized into core transcribed and subtelomeric repressed regions, with some chromosomes containing ESs[7] (Fig. 3A). We obtained ~155 M chromatin contacts, ~118 M intra- and ~37 M inter-chromosomal con-tacts (Fig. 3B and Supplementary Tables 1 and 2). Using a correlation matrix analysis[35], we found that each chromosome contained about two or three compartments with an average length of 1.2 Mb (± 1 Mb, range of ~1-4 Mb), with core transcribed regions forming distinct compartments from silent subtelomeric regions (Fig. 3C and Supple-mentary Fig. 3), analogous to A and B compartments, respectively[1]. Using GC-content, we oriented the core and subtelomeric compart-ments, as subtelomeric regions have lower GC content (~40%) than core regions (~50%). Moreover, using RNA-seq from *T. brucei* blood-stream forms of the SM427 strain[14], we assigned transcribed and silent regions to A and B compartments, respectively (Fig. 3C and

Supplementary Fig. 3). Notably, in *T. brucei*, the organization of A and B compartments correlates well with previously described core and subtelomeric compartments[7] for Mb-long chromosomes, with B compartments also including silent ESs. We then quantified contacts within compartments (intra-chromosomal contacts) and between compartments of different chromosomes, i.e., core vs core (A vs A), core vs subtelomeric (A vs B), and subtelomeric vs subtelomeric (B vs B). We found more intra-chromosomal contacts within subtelomeric compartments (B) than in core compartments (A) (Fig. 3D), consistent with previous observations[7], indicating a more compacted sub-telomeric region. Moreover, we found that subtelomeric compart-ments (B vs B) of different chromosomes co-interacted with higher frequency than core compartments (A vs A) (Fig. 3E), indicating that subtelomeric regions of Mb-long chromosomes are spatially clustered, consistent with our observations by fluorescence in situ hybridizations[21]. The core and subtelomeric chromosome compart-ments were further organized into multiple TADs and loops (Fig. 3C, F, G, Supplementary Fig. 3, and Supplementary Data 6), with the former averaging 280 Kb (±187 Kb, p value ≤ 0.05) and encompassing one to four PTUs (Figs. 3G, 5G), whereas loops averaged 138 Kb (±182 Kb, p-value ≤ 0.05) (Fig. 3F and Supplementary Data 6). The TAD range is consistent with those identified in other organisms[3,36–40]. The inability of previous Hi-C work[7] to identify TADs in *T. brucei* may reflect the low number of intra-chromosomal contacts identified, ~15% of the num-bers reported here. To address whether TADs were an artifact of sequencing alignment[24], we performed Pore-C, which utilizes Oxford nanopore long sequencing and identifies the chromatin ligated regions and multiway contacts[41] (Supplementary Fig. 4). We found TADs with both Hi-C and Pore-C (Supplementary Fig. 4 and Supple-mentary Data 6), indicating that they are not artifacts of alignment since Pore-C reads are 10 times longer (N50 = 1.54 Kb) than the Hi-C reads sequenced (150 bp Illumina reads). Notably, we identified 463 TADs with Pore-C using a 1 Kb resolution matrix (Supplementary Fig. 4, p value ≤ 0.05 and delta ≤0.01, Supplementary Data 6), 6-fold more than the number of TADs identified by Hi-C (77 TADs), even though the Hi-C data contained ~36-fold more contacts than the Pore-C data. The data imply that Pore-C might be a better approach than Hi-C for identifying TADs, perhaps by a combination of long-read alignment and multiway contact detection, also observed in other eukaryotes[42]. A potential role for TADs and loops is the organization of genes around transcription and RNA processing machinery to facilitate/optimize gene expression (Fig. 4A). The data indicate defined levels of intra- and inter-chromosomal organization, as well as enriched inter-chromosomal interactions among silent subtelomeric chromosome compartments.

Our Hi-C and ChIP-seq revealed that RAP1 was markedly enriched at the boundaries of chromosome compartments separating sub-telomeric from core regions (Fig. 3G, H and Supplementary Fig. 3 and 5). Moreover, RAP1 spread over silent subtelomeric regions with a significant enrichment compared to core regions (Fig. 3G, I and Supplementary Figs. 3 and 5). Notably, RAP1 chromatin binding cor-relates with silent subtelomeric (B) compartments (Fig. 3C, G and Supplementary Fig. 3), indicating that it might function to insulate and repress them. KKT2, enriched in centromeres[7,30], as well as SCC1 and H3.V were found at the boundaries of some −but not all−compart-ments (Fig. 3C and Supplementary Fig. 3 and 5), indicating that not all boundaries are centromeric (Fig. 3C and Supplementary Fig. 3 and 5). Other chromatin-associated proteins, such as BDF2, HAT1, HDAC1, and ZCW1, which are enriched at the TSRs of PTUs[33], were also enriched at compartment boundaries, typically flanking RAP1 binding sites, but did not spread over subtelomeric regions (Fig. 3G, H and Supple-mentary Fig. 5). The presence of some of these proteins at the com-partment boundaries could be due to some overlap with TSRs, but they might also have a role in chromatin modification at these regions. The data indicate that chromosome compartment boundaries are

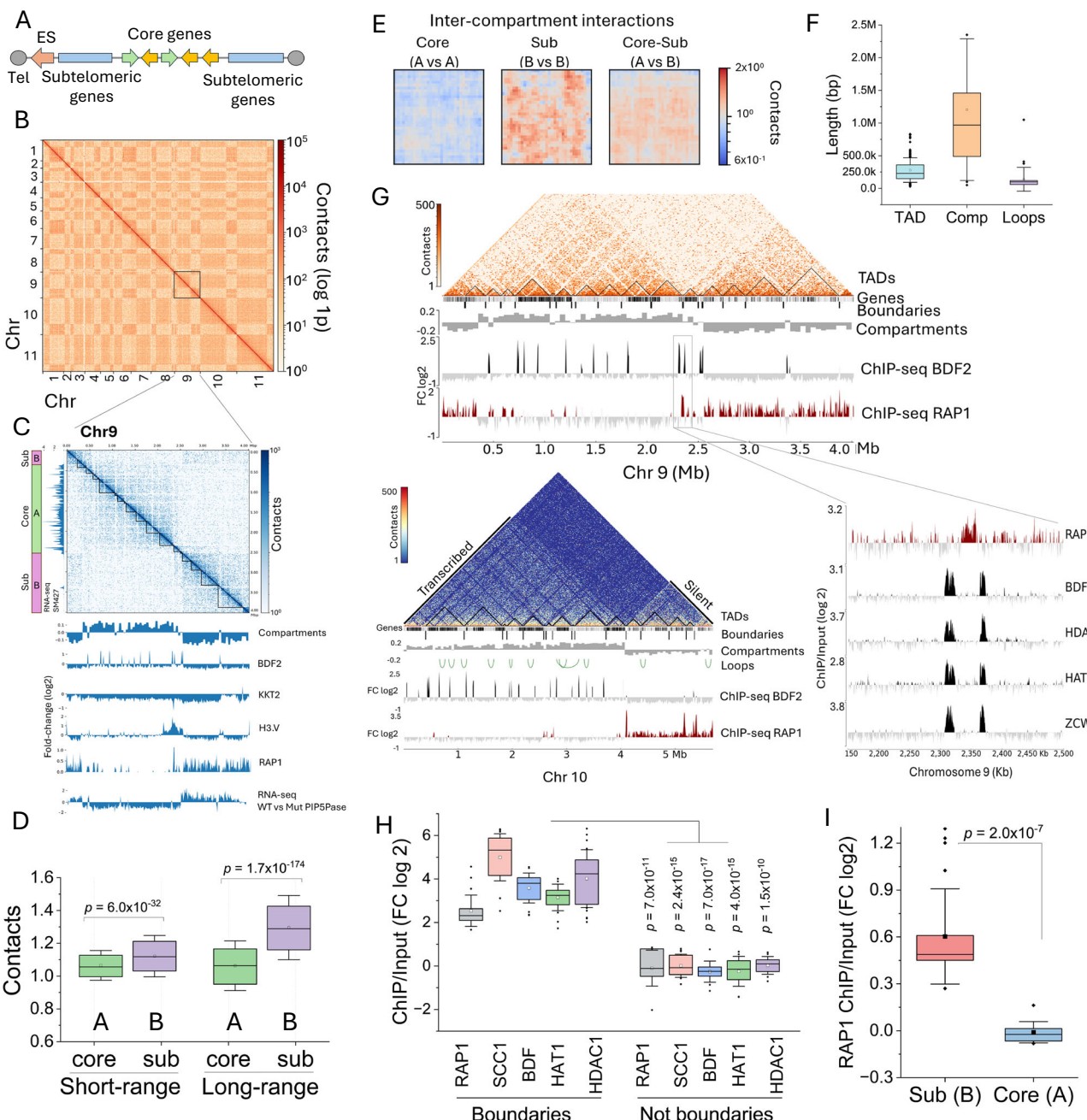

**Fig. 3 | Chromosome spatial organization in *T. brucei* bloodstream forms.**
**A** Diagram of *T. brucei* large chromosomes. Tel, telomere; ES, expression sites. **B** Hi-C heatmap of intra- and inter-chromosomal contacts from chromosomes (Chr) 1 to 11 at 50 Kb resolution. **C** Hi-C heatmap of Chr 9 (top, 10 Kb resolution) with TADs (black lines) indicated. Compartments and ChIP-seq of BDF2, RAP1-HA, KKT2, and H3.V are also indicated (bottom). RNA-seq of *T. brucei* bloodstream forms expressing WT or mutant (Mut) PIP5Pase (D360A/N362A) is also shown. On the left, a diagram of chromosome core and subtelomeres and RNA-seq of *T. brucei* SM427 bloodstream forms indicates transcribed and silent compartments. RNA-seq data in bins per million mapped reads (BPM) from Oxford nanopore sequencing. A mapQ = 30 was used to filter RNA-seq alignments. **D** Average short- and long-range intra-chromosomal interactions from all core (**A**) and subtelomeric (sub, **B**) compartments. Long-range interactions are > 100,000 kb (*N* = 2178), whereas short-range interactions are <100,000 Kb (*N* = 2178). **E** Average of all inter-chromosomal core (**A**) and subtelomeric compartment (**B**) interactions, or the interactions between core and subtelomeric (AB) compartments. **F** Average length of TADs (*N* = 749), compartments (*N* = 30), and loops (*N* = 34) in bp. TADs and loops were obtained from a matrix at 10 Kb resolution. **G** A Hi-C heatmap (10 Kb resolution) of

Chr 9 (top) and 10 (bottom) chromatin contact with compartments and TADs indicated. Compartments were assigned transcribed or silent based on RNA-seq data[13,14]. ChIP-seq data of RAP1-HA and BDF2 are shown below. Inset on the right includes ChIP-seq of RAP1-HA, BDF2, HDAC1, HAT1, and ZWC1 at the Chr9 compartment boundary (-100 Kb segment). **H** Quantification of boundary proteins by ChIP-seq, combining all chromosome compartment boundaries. FC, fold-change of ChIP vs Input. Not boundaries signal was obtained from a 10 Kb region flanking boundaries. *N* = 31, for each protein. **I** ChIP-seq enrichment of RAP1-HA in subtelomeric vs core chromosome compartments. Data shows the average of all 11 Mb-size chromosomes. Core, core transcribed or A compartments; sub, subtelomeric silent or B compartments. *N* = 22. **D**, **E** show contacts in median pixel values[66]. Boxes in **D**, **F**, **H**, and **I** show IQR (25-75%); the center line is the median; the square is the mean; whiskers with minima and maxima shows the ±SDM; and the dots are the outliers. *p* values were calculated using two-sample two-sided *t*-test. Hi-C data is the sum of three biological replicates. RNA-seq and ChIP-seq show the average of three biological replicates. Hi-C matrix used for analysis and heatmaps were balanced with the KR method. Source data are provided as a Source data file.

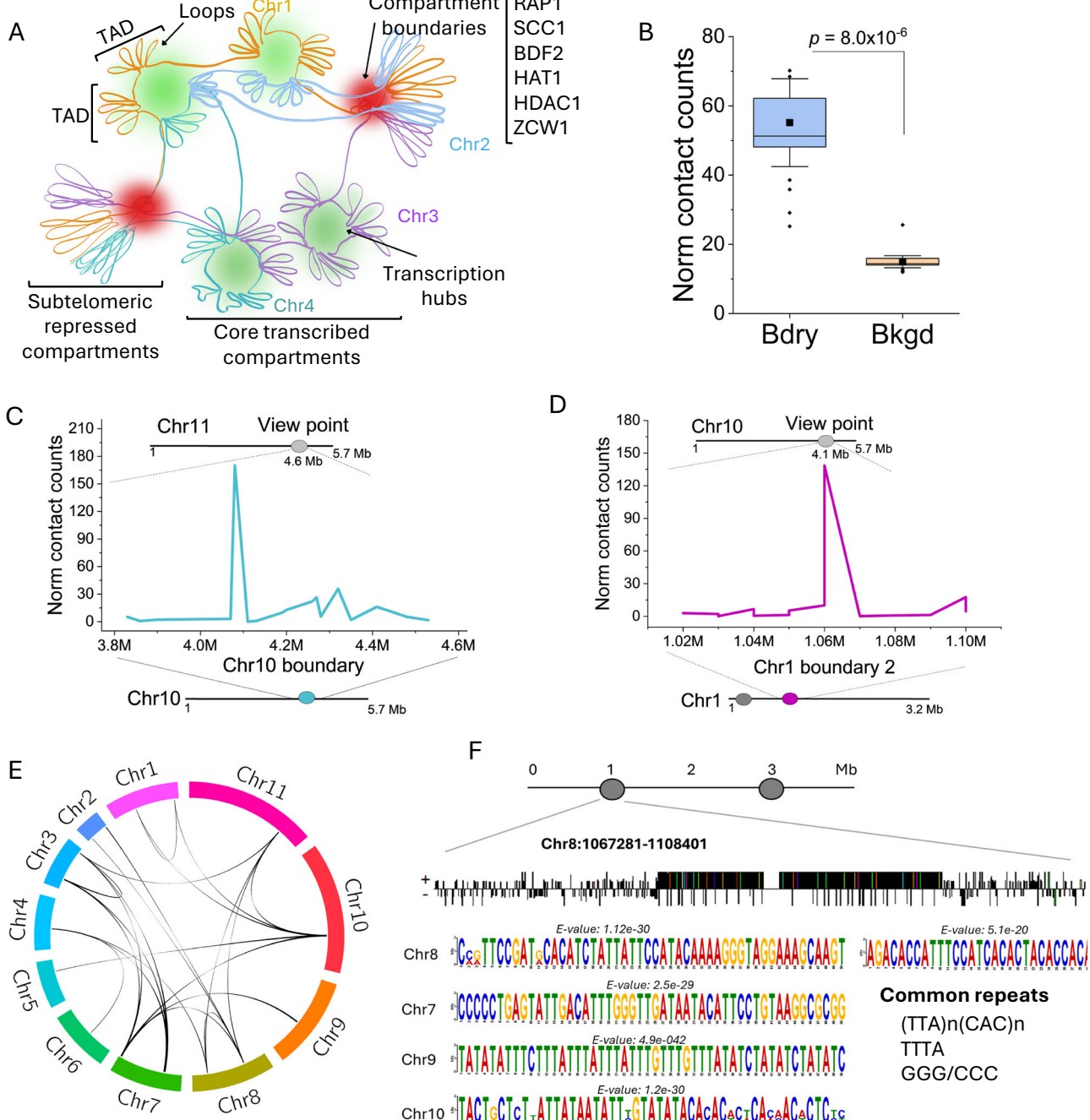

**Fig. 4 | Interactions among chromosome compartment boundaries. A** A model of chromosome organization showing segregation of core transcribed compartments and silent subtelomeric compartments. Compartment boundaries are indicated in red. TADs and loops group PTUs near transcription hubs (in green) that facilitate transcription and RNA processing. Not all chromosome features (centromeres, rDNA, etc.) are indicated. Diagram generated using Inkscape version 1.4 (Inkscape Project, 2024). **B** Quantification of normalized contact counts among boundaries of the same and different chromosomes. Bdry (boundary), mean contact among two boundary regions; Bkgd (background), mean contact among boundaries to all chromosome regions. Boundaries from all 11 Mb-long chromosomes were included in the analysis. Boxes show IQR (25-75%); the center line is the median; the square is the mean; whiskers with minima and maxima are the ± SDM; and the dots are the outliers. *p*-value was calculated using a two-sample two-sided *t*-test. $N = 26$. **C, D** Virtual 4 C plot showing contact boundary between chromosomes 11 and 10 (**C**), and chromosomes 10 and 1 (**D**). **E** Circos plot showing contacting boundaries of the 11 Mb-long chromosomes from the SM427 strain. **F** Diagram of chromosome 8 showing boundaries (grey circles). Below is a diagram of chromosome 8 boundary (~41 Kb, from nucleotides 1,067,281 to 1,108,401). Bars indicate sequence motifs (patterns) identified in the positive (+) or negative (−) strands. Example of conserved sequences and respective E-values are indicated below for a few boundary motifs. See Supplementary Fig. 6 for a complete list of repeated sequences. Common repeat pattern identified in all boundaries is also shown. Hi-C data is the sum of three biological replicates. Source data are provided as a Source data file.

enriched in RAP1, BDF2, HAT1, HDAC1, and ZCW1, and in some cases H3.V and KKT2 (when boundary is centromeric); however, RAP1 is the only protein among those that spreads over the subtelomeric regions. The co-existence of these proteins at the compartment boundaries may explain their cross-links as determined by XLMS, suggesting that they might contribute to insulating the core from subtelomeric chromosome regions. Due to the cross-linking among boundary proteins, we analyzed whether different boundaries also interact. Analysis of the

contact among boundaries from the same or different chromosomes showed higher contact frequency between the boundaries (~10–100 Kb sequence in length) than boundaries contacting any other chromosome regions (Fig. 4B). This is demonstrated by virtual 4C analysis of chromosomes 11 and 10 (Fig. 4C) and chromosomes 10 and 1 (Fig. 4D) showing contact enrichment at the boundaries. The analysis revealed that each boundary interacted with another 3 to 6 boundaries within the same or different chromosomes, e.g., two boundaries of chromosome 8 co-interacted. Still, they also contacted boundaries on chromosomes 11, 7, 4, and 2 (Fig. 4E). Although this arrangement predominates in the single marker 427 strain population (in our laboratory), it may differ among cell lines or throughout the cell cycle. Analysis of boundary sequences revealed an enrichment in $(TTA)_n(CAC)_n$ repeats or sequences with stretches of C/G repeats (Fig. 4F and Supplementary Fig. 6). These sequences resemble those found in the 70 bp repeats within the ESs, but also the AT-rich centromeric regions or telomeric repeats, all of which are binding sites for RAP1[14]. Hi-C and Pore-C reads included only uniquely mapped reads. Analysis of RAP1 ChIP-seq (performed with Oxford nanopore sequencing) with stringent mapping quality (mapQ = 10) retained RAP1 binding to boundaries and subtelomeric regions (Supplementary Fig. 7), indicating that its DNA binding is not an artifact of alignment. The protein composition and chromatin contact suggest that boundaries likely insulate silent from actively transcribed chromosome compartments (Fig. 4A) and may play a role in chromosome organization and subtelomeric VSG silencing.

## Spatial chromatin compartment organization is essential for subtelomeric VSG gene silencing

The data indicate that the PIP5Pase network of chromatin-associated proteins may play a role in boundary formation and chromatin compartment organization. Hence, we postulated that the knockdown of PIP5Pase could disrupt chromatin organization, especially in the silent subtelomeric compartments. We used a conditional null PIP5Pase to determine if its knockdown affects chromatin spatial organization and gene expression within the compartments. We removed the endogenous PIP5Pase alleles and added a tetracycline (tet)-regulatable PIP5Pase, which we showed can be turned on or off in the presence or absence of tet, respectively[13,14,21]. PIP5Pase does not affect cell viability within 24 h[21]; hence, we performed Hi-C in cells expressing PIP5Pase (tet+) or in which we knocked it down for 24 h (tet−) (Fig. 5A and Supplementary Tables 1 and 2). Principal component and matrix analysis showed similar contacts in the Hi-C dataset of SM427 and conditional null cells expressing PIP5Pase (Tet+), but differences with PIP5Pase 24 h knockdown (Tet−) (Supplementary Fig. 8). After PIP5Pase 24 h knockdown, we observed a remarkable decrease (up to 256-fold) in short-range chromatin contacts and a slight increase in long-range contacts, including intra- and inter-chromosomal contacts (Fig. 5B), the latter likely resulting from the disruption of chromosome organization. Analysis of chromosome compartments showed significant disruption of contacts within the compartments (Fig. 5C, D and Supplementary Fig. 9), although boundaries were preserved. Moreover, inter-compartment interactions, i.e., contacts between compartments of different chromosomes, were also affected, primarily by disrupting subtelomeric compartment contacts (B vs B and A vs B) (Fig. 5E). There was a slight increase in interactions among core compartments (A vs A) of different chromosomes after PIP5Pase knockdown (Fig. 5E), likely representing unspecific chromatin interactions due to disruptions of compartment contacts (Fig. 5C) and contact among subtelomere compartments of different chromosomes (Fig. 5E). The data suggest that PIP5Pase has a role in assembling and organizing chromosome compartments, likely via its chromatin protein interaction network. The disruption of intra-chromosomal contacts in knockdown cells significantly affected the formation of TADs (Fig. 5F), resulting in a decrease in TAD numbers or an increase in their

size due to the loss of their boundaries (Fig. 5G), implying that the proteins involved in TAD formation might be part of the PIP5Pase chromatin interaction network. The disruption of TADs by PIP5Pase knockdown further supports the presence of TADs in *T. brucei*. Some chromosomes showed a decrease in the number of loops (Fig. 5G), although loops appear to be less stable than TADs and compartments[25,26,43]. The data indicate a role for PIP5Pase in regulating chromosome-level chromatin spatial organization.

The data suggest that PIP5Pase also coordinates subtelomeric compartment interactions, which may occur via the association of its chromatin protein interaction network at the compartment boundaries and subtelomeric regions. The proteins at the boundaries may be essential for delimiting transcribed from repressed chromatin, thus functioning as a barrier to segregate silent subtelomeric from core transcribed regions. Given RAP1's role as a transcriptional repressor[14,20], its binding at compartment boundaries and over subtelomeric regions (Fig. 3G, H), we posited that PIP5Pase knockdown may disrupt RAP1 binding to compartment boundaries and its spread over subtelomeric compartments, and thus their repression. We analyzed RAP1-HA ChIP-seq in cells expressing a catalytically inactive PIP5Pase (D360A/N362A), which fails to dephosphorylate nuclear PI(3,4,5)P3[13] – an allosteric regulator of RAP1-DNA binding[14]. RAP1 was abolished from chromatin compartment boundaries and significantly decreased from subtelomeric silent (B) compartments in cells expressing catalytically inactive PIP5Pase (Figs. 5G and 6A–C, and Supplementary Fig. 10). Immunofluorescence analysis of HA-tagged RAP1 showed a broader nuclear distribution of RAP1 occupying a 6-fold area increase in cells expressing mutant compared to wildtype PIP5Pase (Fig. 6D, E), perhaps reflecting loss of RAP1-DNA association. Moreover, RNA-seq comparing PIP5Pase catalytic mutant versus wildtype showed that subtelomeric compartment genes were significantly de-repressed, resulting in the expression of subtelomeric VSGs (Fig. 6A, F and Supplementary Figs. 3 and 10). There was a slight decrease in the expression of core genes (Fig. 6A, F and Supplementary Figs. 3 and 10), which might reflect the disruption of core compartment structures, such as TADs and loops, perhaps affecting the efficiency of core gene transcription or RNA processing. Interestingly, PIP5Pase knockdown decreased the frequency of contact boundaries but did not completely disrupt their interactions (Supplementary Fig. 11). The data suggest that contact among boundaries is not necessary to repress silent subtelomeric regions, but it may help demarcate core and subtelomeric compartments. Moreover, the data show that PIP5Pase controls chromatin compartment organization and transcriptional repression of subtelomeric VSG genes via RAP1 association with compartment boundaries and subtelomeric regions.

## Discussion

The spatial organization of chromosomes into core and subtelomeric compartments is essential for silencing subtelomeric genes in *T. brucei*. In this organism, chromosomes are organized into two distinct compartments[7]: transcribed core (A) with genes encoding housekeeping and hypothetical proteins, and silent subtelomeric (B) with primarily VSG genes. Using XLMS, we uncovered chromatin-associating factors contributing to this chromosome spatial organization, specifically proteins associated with compartment boundaries, including RAP1, HDAC1, HAT1, BDF2, ZCW1, and SCC1. We show that different compartment boundaries co-interact, likely helping to insulate silent subtelomeric regions from the transcribed core chromosome regions. Subtelomeric compartment silencing depends on RAP1, which spreads over these regions, silencing hundreds of VSG genes. Subtelomeric gene silencing also depends on PIP5Pase activity, which controls contacts within compartments and RAP1 binding to DNA. The data point to a regulatory mechanism that spatially segregates transcribed and repressed chromatin and represses hundreds of VSG

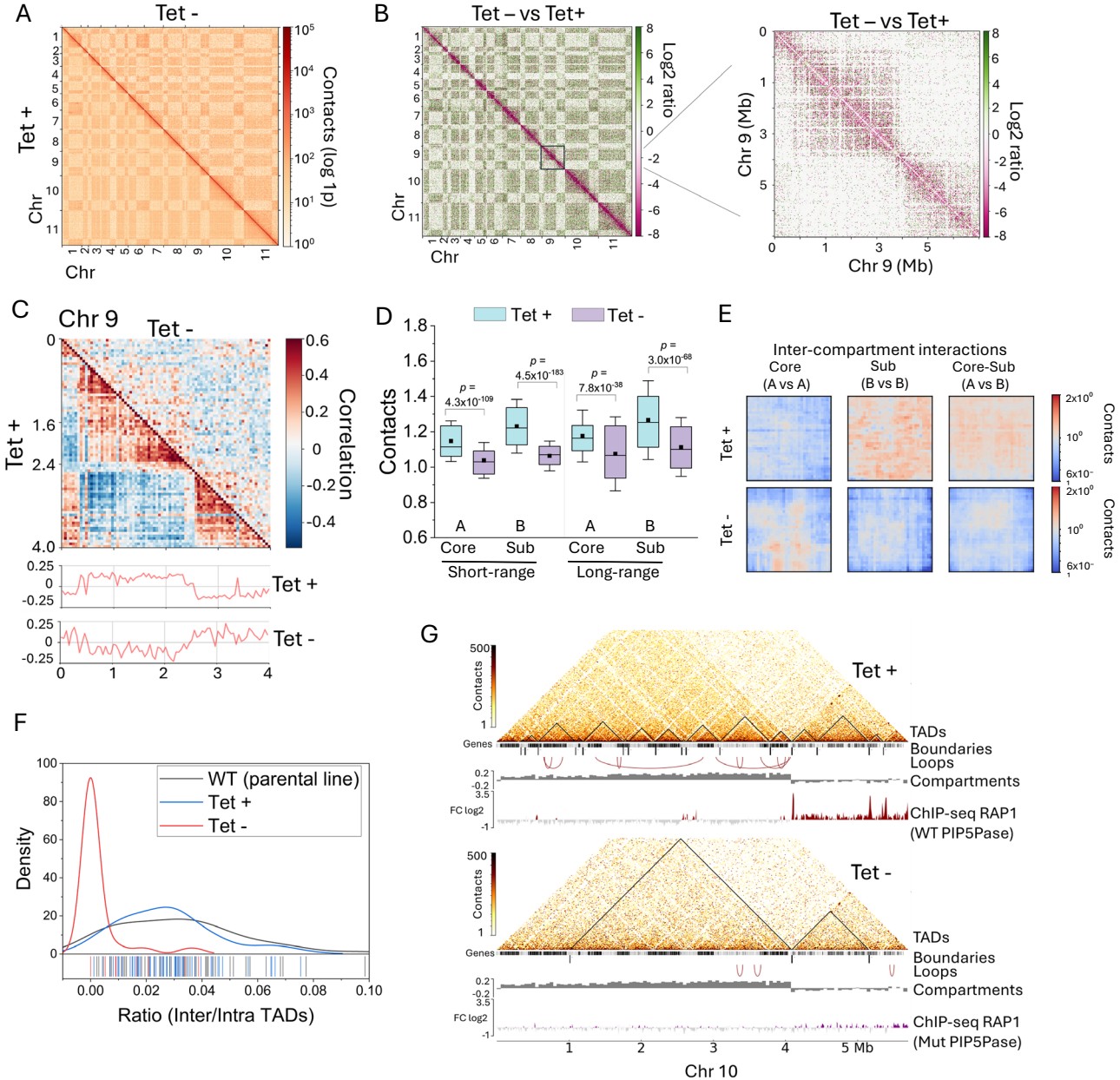

**Fig. 5 | PIP5Pase regulates chromatin spatial organization. A** Hi-C heatmap of Chr 1-11 contacts *in T. brucei* expressing PIP5Pase (Tet +, lower left) or after 24 h knockdown (Tet−, upper right) at 50 b resolution. Tet, tetracycline. **B** Hi-C heatmap of Chr 1-11 log2 ratio of contacts from PIP5Pase 24 h knockdown (Tet-) versus PIP5Pase expressing cells (Tet+) at 50 Kb resolution. On the right, Hi-C heatmap of Chr 9 log2 ratio of chromatin contacts from Tet− versus Tet+ cells at 10 Kb resolution. **C** Hi-C heatmap of Chr 9 chromatin contacts at 10 Kb resolution from cells expressing PIP5Pase (Tet+, lower left) vs 24 h knockdown (Tet−, upper right). Compartment scores are shown below. **D** Average short- and long-range intra-chromosomal contacts from all core (**A**) and subtelomeric (**B**) compartments in cells expressing PIP5Pase (Tet+) or after 24 h knockdown (Tet−). Long-range interactions are >100,000 kb (N = 2178), whereas short-range interactions are <100,000 Kb (N = 2178). Boxes show IQR (25–75%); the center line is the median; the square is the mean; whiskers with minima and maxima are the ±SDM. *p*-values were calculated using two-sample two-sided *t*-test. **E** Average of all inter-

chromosomal core (**A**) and subtelomeric (**B**) compartment contacts in cells expressing PIP5Pase (Tet+) or after 24 h knockdown (Tet−). **F** Ratio of contacts between (Inter) and within (Intra) TADs in cells expressing PIP5Pase (Tet +) or after 24 h knockdown (Tet−). Density, kernel density distribution of the data. X-axis bars show ratio data distribution. WT, SM427 cells (N = 278). **G** Heatmap of Chr 10 chromatin contacts (10 Kb resolution) from cells expressing PIP5Pase (Tet +) or after 24 h knockdown (Tet−). TADs (from 10 Kb resolution matrices), TAD boundaries, loops, and compartments are indicated. RAP1-HA ChIP-seq from cells exclusively expressing PIP5Pase (WT) or its catalytic inactive mutant (Mut, D360A/N362A) is also shown. **D**, **E** show contacts in median pixel values[66]. ChIP-seq shows the average of three biological replicates. Hi-C data is the sum of three biological replicates for Tet+ and three biological replicates for Tet−. Hi-C matrices used for analysis and graphing were balanced with KR method and normalized to the smallest matrix for comparisons between groups. Source data are provided as a Source data file.

genes. Furthermore, it shows a role for phosphoinositides in the control of genome spatial organization.

In addition to compartments, we show that *T. brucei* chromosomes are further sub-organized into TADs and loops with length ranges correlating with other eukaryotes[2,3,36–40]. This organization is

further arranged around polymerase-specific hubs that coordinate the transcription of mRNAs, rRNAs, and tRNAs[24]. This chromosome spatial organization likely reflects the cell's need to control transcription, which in some organisms occurs by placing co-regulated genes, promoters, and enhancers in proximity[2,6]. *T. brucei* lacks RNA polymerase

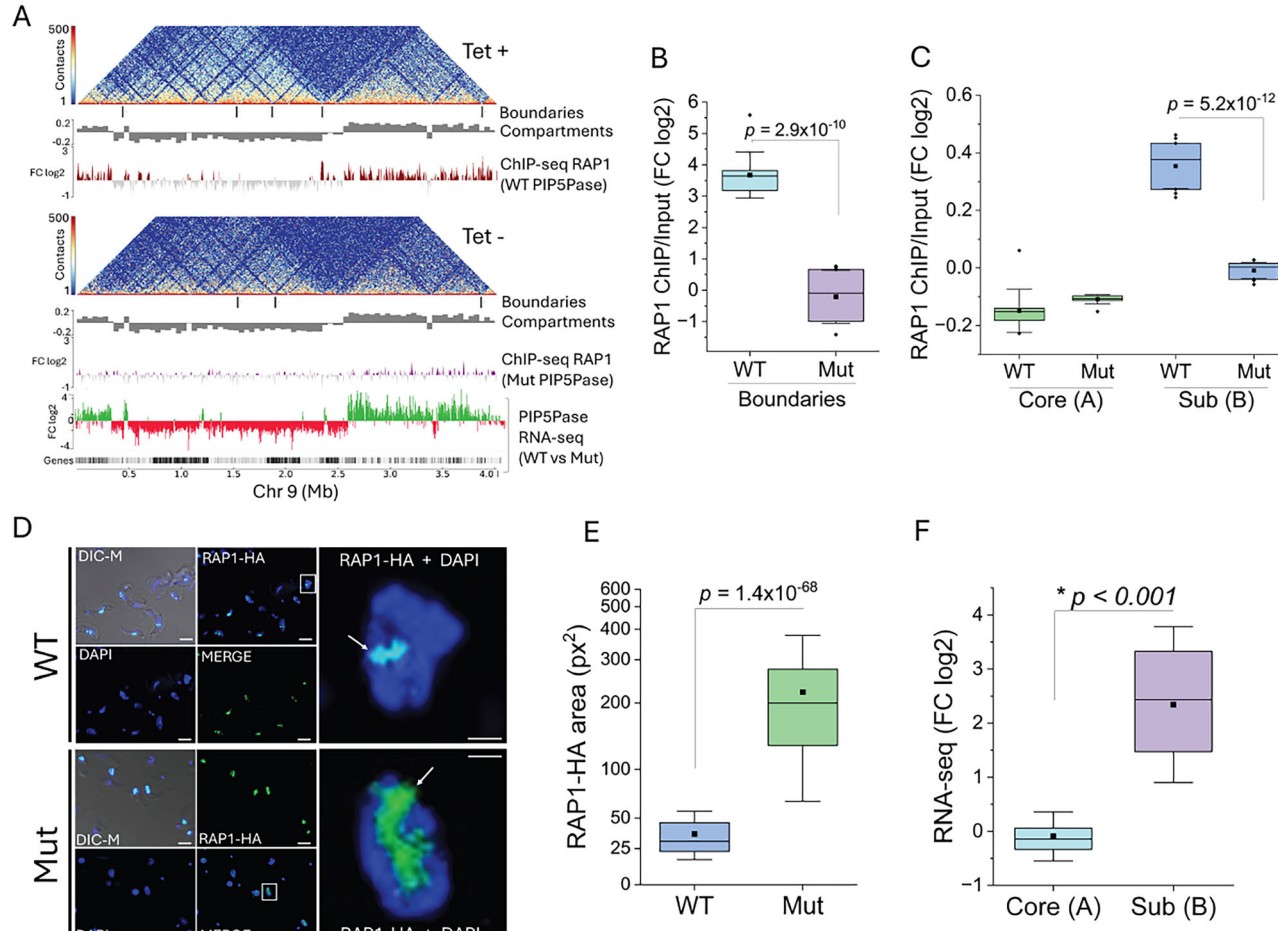

**Fig. 6 | PIP5Pase and RAP1 repress VSG-rich subtelomeric compartments.**
**A** Heatmap of Chr 9 chromatin contacts at 10 Kb resolution in cells expressing PIP5Pase (Tet +) or after its 24 h knockdown (Tet−). Tet, tetracycline. Below the heatmap are TAD boundaries, compartments, RAP1-HA ChIP-seq in cells exclusively expressing PIP5Pase (WT) or its catalytic inactive mutant (Mut, D360A/ N362A), and RNA-seq log2 fold-change (FC) from cells exclusively expressing PIP5Pase (WT) versus its catalytic inactive mutant (Mut). Green, upregulated genes; red, downregulated genes. **B**, **C** RAP1-HA ChIP-seq enrichment in compartment boundaries (**B**, $N = 22$) and subtelomeric and core compartments (**C**, $N = 22$) in cells exclusively expressing WT or Mut PIP5Pase. FC, fold-change of RAP1-HA ChIP-seq vs Input. **D** Confocal microscopy of HA-tagged RAP1 (green) in *T. brucei* exclusively expressing WT or Mut PIP5Pase for 24 h. DNA stained with DAPI (blue). Bar = 10 μm, inset bar = 2.5 μm. DIC-M, merged differential interference contrast and fluorescence images. DAPI, 4′,6-diamidino-2-phenylindole.

The image is representative of three biological replicates. **E** Quantification of the nuclear area occupied by RAP1-HA (as shown in D). $N = 458$ nuclei. px2, square pixel. **F** Log2 fold-change (FC) in the expression of genes from subtelomeric and core compartments in cells exclusively expressing WT or Mut PIP5Pase for 24 h. Data show transcript levels of genes within all core ($N = 8572$ genes) or all subtelomeric ($N = 4416$ genes/pseudogenes) chromosome compartments. Boxes in **B**, **C**, **E**, and **F** show IQR (25-75%); the center line is the median; the square is the mean; whiskers with minima and maxima are the ± SDM. *p*-values were calculated using two-sample two-sided *t*-test. * in F indicates *p* value is too low, nearly 0. RNA-seq and ChIP-seq show the average of three biological replicates. Hi-C data are the sum of three biological replicates for Tet+ and three biological replicates for Tet−. Hi-C matrices used for analysis and graphing were balanced with KR method and normalized to the smallest matrix for comparisons between groups. Source data are provided as a Source data file.

II transcriptional control, and genes within PTUs of core chromosome regions are co-transcribed[44]. We show that each TAD encompasses one to four PTUs. A possible role for TADs and loops in *T. brucei* might be to organize subsets of PTUs around transcriptional hubs (Fig. 4A). This arrangement could provide efficient transcription and RNA processing by grouping and recycling the transcription machinery around contact regions. Previous work also suggested TSRs may form distinct interchromosomal transcription hubs, which may facilitate RNA polymerase II transcription[24]. Proteins present at TSRs, such as BDF2, HAT1, and HDAC1, may help organize chromatin contact in these regions and facilitate recruitment of the transcription machinery. This model is consistent with the decrease in the expression level of core genes after PIP5Pase knockdown, which we showed disrupted core compartment contacts, TADs, and loops. Muller and colleagues did not report TADs in *T. brucei*[7]. However, Diaz-Viraque and colleagues, analyzing the same dataset, found them in *T. brucei*, but also in *T. cruzi*[45], describing them

as chromatin folding domains (CFD). We use the term TAD instead of CFD in strict accord with the methodology and algorithms used, as well as its broad use in other eukaryotes. Micro-C performed with high contact numbers also revealed TADs, but with low confidence[24], perhaps because of extensive micrococcus nuclease digestion of the typically open *T. brucei* chromatin. We demonstrate that Hi-C performed with high sequence depth reveals TADs, but Pore-C seems superior even with lower sequencing depth. The differences may arise from the length of Pore-C reads−with longer reads mapping better to low complexity genome sequences− and the method to identify contacts, as Pore-C reveals with precision ligated DNA sequences[41] and allows the identification of complex multiway contacts that occur within TADs[46]. Others have also reported a higher number of TADs identified by Pore-C when compared to Hi-C[42], consistent with our observations in *T. brucei*. The findings that PIP5Pase knockdown also disrupted the TADs further support the existence of TADs in *T. brucei*,

as knockdown of this protein shows that TAD formation depends on a specific chromatin regulatory factor. Whether phosphoinositides such as PIP5Pase substrates, PI(4,5)P2, and PI(3,4,5)P3[13,21], play a role in TAD organization in other eukaryotes is unknown. A possible role for PIP5Pase in regulating TAD assembly might relate to its association with the SLF1 subunit of the cohesin complex, as identified by XLMS. An ortholog of DNA-binding protein CCCTC-binding factor (CTCF), which is involved in TAD and loop formation in some eukaryotes[47], has not yet been identified in *T. brucei*, but processes analogous to what occurs in other eukaryotes might be engaged in TAD and loop formation in this parasite, perhaps with other DNA-binding proteins.

In contrast to transcribed chromosome core regions, subtelomeric regions are rich in VSG genes and are transcriptionally inactive. We found specific chromatin-binding proteins at the boundaries between core (A) and subtelomeric (B) compartments, such as RAP1, BDF2, HAT1, HDAC1, SCC1, and ZCW1. Not all boundaries contain all proteins, but a combination of them, with RAP1 present in nearly all of them. If a boundary is at the centromere, then KKT2, RAP1, and SCC1 are present. Some of these factors (BDF2, HAT1, HDAC1, ZCW1) are also present at TSRs[33], some of which are within the boundaries. These proteins are typically neighbouring RAP1. The presence of these factors at the compartment boundaries could coincide with the TSRs, or perhaps they have a direct role in chromosome compartment boundaries associated with their function in chromatin acetylation or methylation. The boundaries and proteins associated thereof might function to delimit transcribed and repressed states of the chromatin. We show that the boundaries co-interact, and perhaps some of these proteins identified by XLMS might be mediating their contacts. The interactions among the boundaries might be stably formed every cell cycle or be dynamic. Since we observed different combinations of boundary contacts in the single-marker 427 strain (parental cell line) and the PIP5Pase conditional null cells, the arrangement of boundary interactions is unlikely to be fixed and perhaps varies between cells, as the Hi-C represents a populational analysis. Although PIP5Pase knockdown decreased contact frequency at the compartment boundaries, the contacts were not completely disrupted, implying that boundary contacts are not required for silencing subtelomeric regions. They may play a role in demarcating transcribed and repressed chromatin regions. The removal of RAP1 from boundaries and subtelomeric compartments after PIP5Pase inactivation concurred with disruption in chromatin contacts within compartments and transcriptional activation of subtelomeric genes, indicating that spatial compartment organization and both proteins are required to silence subtelomeric chromatin. The higher frequency of interactions among subtelomeric compartments of various chromosomes is consistent with observations by FISH[21,48], and might favour the association of factors involved in their repression, such as RAP1, and perhaps contribute to co-occupancy of physical spaces within the nucleus. The *T. brucei* genome might have evolved physical and spatial segregation of silent and active genes to compensate for the lack of transcriptional control.

The proteins associated with the boundaries were identified by XLMS of immunoprecipitated PIP5Pase. The XLMS data showed that the PIP5Pase interaction network comprises proteins involved in chromatin organization, transcription, splicing, DNA repair and recombination, and others. The cross-linking of these proteins within a short ~20–30 Å range might have resulted from their direct binding and/or the spatial proximity of their chromatin binding sites. For example, factors involved in transcription, DNA replication, or chromatin organization may co-exist associated with the same chromosome regions, but play distinct roles. This could, for example, reflect interactions around boundaries, TADs, and transcription hubs. Notably, the cross-link indicates the proximity of proteins and not their direct interaction. Conversely, the lack of cross-link does not imply a lack of interaction due to the approach's stringency, requiring two cross-linkable amino acids within ~20-30 Å[28], as we demonstrated for

PIP5Pase and RAP1. The immunoprecipitation of PIP5Pase (373 peptides) pulled down RAP1 (241 peptides); however, we did not detect in vivo cross-links between these proteins, which might be due to the lack of cross-linkable amino acids within the ~20–30 Å constraints of DSS. Nevertheless, the two proteins co-immunoprecipitate as reported before[13,21,33], and we confirmed their direct interactions by SPR. The XLMS provides a map of the interacting domains of several bona fide nuclear proteins in this organism, indicating how different protein complexes and chromatin proteins might co-interact and, notably, how the spatial organization of the genome arranges these proteins in proximity for their activity.

Chromosome compartments are likely formed every cell cycle after DNA replication. PIP5Pase might function to control RAP1 binding to compartment boundaries and subtelomeric regions during compartment assembly through/after DNA replication, and the failure of RAP1's association with these regions may affect compartment assembly and their repression. This binding is dependent on PIP5Pase activity, and thus, accumulation of PI(3,4,5)P3, which we showed binds to RAP1 N-terminal and displaces it from DNA[14]. The mechanistic dependence on phosphoinositides for RAP1 binding to compartment boundaries and subtelomeric heterochromatin silencing has not been observed elsewhere. In yeast, repression of heterochromatin requires the silent information regulator complex[49,50], which is not involved in VSG silencing in *T. brucei*[51]. It is unknown what drives the association of other factors to the boundaries, but the RAP1 association may precede them, demarking the region for their binding. We showed that disruption of PIP5Pase activity also results in VSG switching by transcriptional switching among ESs and by recombination of subtelomeric VSG genes within ESs[14,52]. We found here that subtelomeric compartments of multiple chromosomes co-interact, which may help isolate the silent subtelomeric genes. Disruption of this chromosome organization by PIP5Pase knockdown increases unspecific contact among various chromosomes, including with telomeric ESs, which may favour VSG gene recombination. The disruption of chromatin organization by PIP5Pase knockdown was also observed by FISH[21]. Regulation of these processes implies that they occur in regions containing phosphoinositides. Phosphoinositides are synthesized in the endoplasmic reticulum and Golgi and redistributed to other cellular compartments, including the nucleus, where they are modified[21,53]. PI(3,4,5)P3 association with RAP1 might result from this protein's proximity to nuclear membranes through interactions with nuclear lamina proteins[13], such as NUP-1, which knockdown also affects subtelomeric gene silencing[48]. Alternatively, these metabolites may associate with the chromatin via nucleoplasmic lipid droplets containing phosphoinositides[54]. In any case, the data indicate a phospholipid signalling control of chromatin spatial organization. PIP5Pase is essential for *T. brucei* infection[15], and considering its regulatory roles, it could be explored as a drug target.

Our results are consistent with a model in which *T. brucei* chromosomes evolved to physically and spatially segregate transcriptionally active from repressed genes via chromosome organization into transcribed and repressed compartments. Moreover, it shows that RAP1, along with other chromatin-associating factors such as BDF2, HDAC1, HAT1, etc., marks the compartment boundaries, whereas RAP1 also spreads over subtelomeric regions and represses subtelomeric VSG genes. Given that RAP1 also associates with silent telomeric ESs[13,14], it may function as a general transcriptional repressor in this organism. The regulation of chromosome topology by PIP5Pase indicates an epigenetic role for phosphoinositides in controlling genome spatial organization, a process that may be conserved in other eukaryotes.

## Methods
### Cell culture and cell lines
*T. brucei* Lister 427 strain single-marker (SM427) was obtained from Dr. Ken Stuart's laboratory (Seattle Children's). *T. brucei* SM427 and

derived cell lines were maintained in HMI-9 medium supplemented with 10% (vol/vol) fetal bovine serum (FBS) at 37 °C with 5% CO$_2$. *T. brucei* expressing the C-terminally V5-tagged PIP5Pase or HA-tagged RAP1 were generated by transfection of pMOTag2H or pMOTag3V5 into one of their respective alleles, as previously described[13], and maintained in the presence of 0.1 μg/mL of puromycin. Conditional null PIP5Pase in bloodstream forms[21], or modified lines to exclusively express wildtype or mutant (D360A/E362A) PIP5Pase[13] were maintained at 10 μg/mL of zeocin and 0.1 μg/mL of nourseothricin N-acetyl transferase. Conditional null PIP5Pase in procyclics[14] were maintained at 27 °C in SDM-79 (Wisent) medium containing 10% FBS and penicillin/streptomycin (Thermo Fisher Scientific), supplemented with 15 μg/mL of G418, 25 μg/mL of hygromycin, and 2.5 μg/mL of zeocin. Tetracycline at 0.5 μg/mL was added to CN PIP5Pase cultures to induce PIP5Pase expression.

### In vivo chemical cross-linking of *T. brucei* bloodstream forms

*T. brucei* cells were grown in five litres of HMI-9 medium as described above. The cells were washed five times in phosphate-buffered saline (100 mM sodium phosphate dibasic anhydrous, 100 mM sodium phosphate monobasic anhydrous, 145 mM sodium chloride) supplemented with 6 mM glucose (PBS-G). Cells were centrifuged at 4500 × g then resuspended in 500 ml of PBS-G. DSS (ProteoChem) dissolved in dimethylsulfoxide (DMSO) (Sigma) was added at 0.25 mM to the cell suspension. The mix was incubated for 10 min with gentle shaking at 37 °C. The reaction was quenched by adding 50 mM glycine, pH 9, and incubated for 15 min at room temperature (RT). Cells were spun down at 3500 × g, the pellet resuspended in 10 ml of lysis buffer (50 mM Tris, 150 mM NaCl, 1% Triton X-100, 0.5% sodium deoxycholate, 0.1% NP-40, 2x protease inhibitor cocktail (Roche), pH 8.0), and then incubated for 30 min, gently rotating at 4 °C. Lysate was centrifuged at 14,000 × g for 20 min at 4 °C, and soluble proteins were collected from the supernatant. Proteins were analyzed in 10% SDS-PAGE and by Western blot for PIP5Pase-V5 expression before immunoprecipitations.

### Immunoprecipitation of PIP5Pase from cross-linked cells

Monoclonal antibodies (mAbs) α-V5 (ABclonal Inc., catalog # AE017) were cross-linked to protein G magnetic beads (Cytiva) as previously described[13]. Fifty microliters of Protein-G-α-V5 mAbs (20 μg of α-V5) were incubated with 10 mL of protein lysate and incubated overnight, gently rotating at 4 °C. Afterward, the mixture was washed in wash buffer (50 mM Tris-HCl, 300 mM NaCl, 0.1% NP40, 2x protease inhibitor cocktail (Roche), pH 8.0) using a magnetic stand. The proteins were eluted twice (200 μl each) in 6 M urea/100 mM glycine, pH 2.9. Ten percent of the eluate was collected for SDS-PAGE and Western analysis, and the remaining portion was kept at −80 °C for mass spectrometry analysis. Eleven biological replicates were performed. For one experiment, PIP5Pase-V5 was immunoprecipitated from non-cross-linked cells, and cross-link performed with the immunoprecipitated material using 0.25 mM DSS for 10 min with gentle shaking at 37 °C. The reaction was quenched by adding 50 mM glycine, pH 9, and incubated for 15 min at RT.

### Protein digestion and mass spectrometry

Protein digestion was adapted from ref. 55. A total of 15 μg of protein was used for digestion. Briefly, six volumes of cold acetone (−20 °C) were added to immunoprecipitation eluate, vortexed, and incubated for 1 h at −20 °C for protein precipitation. Afterward, protein samples were centrifuged at 14,000 × g for 30 min at 4 °C. The acetone was decanted and air-dried for 15 min, and pellets were resuspended in 10 μl of 6 M urea. Then, 5 μl of reducing solution (10 mM DTT in 100 mM NH$_4$CO$_3$) was added, and samples were vortexed and incubated for 1 hour at room temperature. Samples were spun down (3000 × g for 15 s), 3 μl of alkylation solution (50 mM iodoacetamide in 100 mM NH$_4$CO$_3$) was added, and samples were incubated in the dark

for 30 min at RT. Next, 3 μl of reducing solution was added to neutralize the reaction. Lys-C/trypsin mix (Promega) was added to the protein in a ratio of 1:50 (protein: protease amounts in μg), mixed, and incubated for 4 h at 37 °C. The sample was diluted six-fold with 50 mM Tris-HCL (pH 8.0) to bring the concentration of UREA to 1 M, and the reaction incubated overnight at 37 °C. The reaction was terminated by adding 13.06 M trifluoroacetic acid to a final concentration of 1%. Any particulate material was removed by centrifugation at 14,000 × g for 10 min. The supernatant was analyzed in 10% SDS/PAGE to verify digestion. The samples were analyzed in Thermo Orbitrap Fusion LC-MS at the McGill University Proteomics Centre. Briefly, peptides were solubilized in 0.1% aqueous formic acid, loaded onto a Thermo Acclaim Pepmap (Thermo, 75 μM ID X 2 cm C18 3 μM beads) precolumn and then onto an Acclaim Pepmap Easyspray (Thermo, 75 μM × 15 cm with 2 μM C18 beads) analytical column separation using a Dionex Ultimate 3000 uHPLC at 230 nl/min with a gradient of 2–35% organic (0.1% formic acid in acetonitrile) over 2 h. Peptide analysis was done using a Thermo Orbitrap Fusion mass spectrometer operating at 120,000 resolution (FWHM in MS1) with HCD sequencing (15,000 resolution) at top speed for all peptides with a charge of 2+ or greater.

### Cross-link mass spectrometry data analysis

Peptide and protein identification were done using the Trans Proteomic Pipeline v7.1.0 (http://www.tppms.org/)[56]. Thermo RAW files were first converted to mzML with MSConvert (ProteoWizard release 3.0.24121), and cross-linked peptides were identified using Kojak version 2.0.3[57] using the DSS 136.068 mass for two peptide cross-links, and 156.07864 for mono links, and remaining default parameters except for top_count adjusted to 50 min_peptide_mass to 600, max_peptide_mass to 8000 ppm_tolerance_pre to 25, MS1_resolution and MS2_resolution adjusted to 60,000 and 50,000, respectively, min_peptide_score to 0.5 min_spectrum_peaks to 12 and max_spectrum_peaks to 0. The search was done against *T. brucei* 927 strain reference genome v9.0 predicted protein sequence database. A false discovery rate (FDR) threshold of 0.01, with a precursor mass tolerance of 10 ppm and two missed cleavages allowed, was used. Cross-linking data was validated using Percolator release rel-3-07-01[57]. Protein interactions were visualized using Cytoscape 3.9.1[58], ProXL version 2.4.2[59], and XlinkCyNET version 1.5.3[60]. Enrichment analysis was calculated using total obtained peptides comparing anti-V5 monoclonal antibodies (ABclonal) immunoprecipitations from *T. brucei* bloodstream forms expressing PIP5Pase-V5 (9 biological replicates) or cells expressing non-tagged PIP5Pase (4 biological replicates) to obtain log2 fold-change and *p*-value using a two-sample two-sided t-test.

### Western blotting

Western blot analysis was performed as previously described with modifications[13]. Briefly, lysates of *T. brucei* were prepared in lysis buffer containing 50 mM Tris-HCl, pH 8.0, 300 mM NaCl, 0.1% NP40, 1% Triton-X-100, 2x protease inhibitor cocktail (Roche), and 0.5% sodium deoxycholate. Cleared lysate was mixed in 4x Laemmli loading buffer containing with 2.85 M β-mercaptoethanol, and heated for 5 min at 95 °C. Proteins were resolved in 10% gels and transferred to a polyvinylidene difluoride (PVDF) membrane. The membrane was probed for 2 h at RT (or overnight at 4 °C) with mAbs α-HA (ABclonal, catalog # AE065) or mAb α-V5 (ABclonal, catalog # AE017) diluted 1:5000 in 10% skimmed milk (Bio-Rad) dissolved in PBS 0.05% Tween 20. The membranes were incubated with goat anti-mouse IgG (H + L)-HRP 1:2000 (Life Technologies, catalog # 31430) and developed by chemiluminescence using a GelDoc Imaging System (Bio-Rad).

### Immunofluorescence assay

*T. brucei* were grown to mid-log phase, washed three times in PBS-G by centrifugation at 3500 × g for 5 min, fixed with 2% paraformaldehyde

(Electron Microscopy Sciences, PA) in PBS, and adhered to poly-l-lysine-treated 2-mm cover slips (Fisher). Cells were permeabilized with 0.2% NP-40 (Sigma Aldrich) in PBS for 10 min and blocked for 1 h at RT in blocking buffer (10% nonfat dry milk diluted in PBS). Cover slips were incubated for 2 h at RT in mAb α-HA 3F10 (Invitrogen, catalog # 26183) at a ratio of 1:500 diluted in blocking buffer, followed by goat α-mouse IgG (H + L)−Alexa Fluor 488 (Thermo Fisher Scientific, catalog # A11001) diluted 1:1000 in 10% nonfat dry milk diluted in PBS. Slides were mounted with Fluoromount G mounting medium with 4′,6-dia-midino-2-phenylindole (DAPI) (Biotium). Images were acquired with Zeiss Confocal Laser Microscope and analyzed with Zen Microscopy software (Zeiss).

## Hi-C library and sequencing

Hi-C was performed in *T. brucei* bloodstream forms, single marker 427 (SM427) or conditional null PIP5Pase. PIP5Pase was knocked down for 24 h (tet−) and compared to cells expressing PIP5Pase (tet+). $3 \times 10^8$ cells were fixed in 1% paraformaldehyde in PBS for 10 min at RT and quenched with 0.2 M glycine for 5 min. Cells were washed in PBS by centrifugation at $4000 \times g$ for 10 min. Pellets were processed for Hi-C adapted from ref. 46. Briefly, cells were lysed in lysis buffer (10 mM Tris-HCl, pH 8, 10 mM NaCl, 0.2% NP-40, and 1x protease inhibitor cocktail from Roche), and nuclear pellets were extracted and digested in NEB Buffer 2 with 100 units of *Mbo* I (New England Biolabs) at 37 °C overnight with rotation. After *Mbo* I heat inactivation at 62 °C for 20 min, DNA ends were filled with 1 mM biotinylated dATP (Thermo Fisher Scientific) and 10 mM dCTP, dGTP, dTTP with 50 units DNA polymerase I Klenow fragment (New England Biolabs) for 4 h at 23 °C. DNA ends were ligated with 4000 units of T4 DNA ligase (New England Biolabs) at 16 °C overnight, shaking at 80 rpm. Lysate was digested with 1.1 mg of proteinase K (New England Biolabs) at 63 °C for 4 h, and DNA was extracted by phenol:chloroform:isoamyl alcohol (25:24:1, v/v) method. The DNA was sonicated using a Covaris M220 ultrasonicator (50 peak incidence power, 20% duty factor, 200 cycles per burst, time 180 s). DNA fragments ranging from 100 to 450 bp were size selected from 1% agarose gel/TBE, end-repaired, and A-tailed with Ultra II End Prep kit (New England Biolabs), and DNA-sequencing library prepared with NEBNext oligos for Illumina (New England Biolabs). Libraries were sequenced at GenomeQuebec using an Illumina NovaSeq sequencer for paired-end 150 bp sequencing. A total of ~3.8 billion reads was obtained.

## Pore-C library and sequencing

Pore-C was performed in *T. brucei* bloodstream forms single marker 427. $1.4 \times 10^9$ cells were fixed in 1% paraformaldehyde in PBS for 10 min at RT, rotating at 150 rpm, and quenched with 0.2 M glycine for 5 min. Cells were washed in PBS by centrifugation at $4000 \times g$ for 10 min. Pellets were processed for Pore-C adapted from[46]. Briefly, cells were resuspended in lysis buffer (10 mM Tris-HCl pH 8, 10 mM NaCl, 0.2% NP-40, and 1x protease inhibitor cocktail from Roche) at 4 °C for 30 min with rotation of 80 rpm. Cells were spun down at $2500 \times g$ for 5 min, and the pellets were resuspended in 100 μL of 0.05% sodium dodecyl sulfate (SDS) and incubated at 62 °C for 10 min. After, 285 μL of water and 50 μL of 10% Triton X-100 were added and incubated for 15 min at 37 °C, rotating at 80 rpm. Cells were spun down as above, and pellets digested in NEB Buffer 2 with 100 units of *Mbo* I (New England Biolabs) at 37 °C[52] overnight with 80 rpm rotation. After, *Mbo* I was heat-inactivated at 62 °C for 20 min, and DNA ends were filled with 1 mM biotinylated dATP (Thermo Fisher Scientific) and 10 mM dCTP, dGTP, dTTP with 50 units DNA polymerase I Klenow fragment (New England Biolabs) for 4 h at 23 °C. DNA ends were then ligated with 4000 units of T4 DNA ligase (New England Biolabs) at 16 °C overnight, shaking at 80 rpm. Lysate was digested with 1.1 mg of proteinase K (New England Biolabs) at 63 °C for 4 h, and DNA was extracted by

phenol:chloroform:isoamyl alcohol (25:24:1, v/v) method. The DNA was pulled down using 40 μg streptavidin beads in 2X biotin-binding buffer (10 mM Tris-HCl, pH 8.0, 1 mM EDTA, and 2 M NaCl) after 1 h incubation, rotating at 80 rpm at RT. Then, washed three times in wash buffer (5 mM Tris-HCl, pH 8.0, 0.5 mM EDTA, 1 M NaCl, 0.05% Tween 20) for 2 min at 55 °C, rotating at 300 rpm. DNA was resuspended in water. Samples were treated with 1.1 mg of proteinase K (New England Biolabs) at 63 °C for 4 h, and DNA was extracted by phenol:chloroform:isoamyl alcohol (25:24:1, v/v) method. DNA was cleaned up with NucleoMag® B-Beads (Macherey-Nagel) at 0.8:1 beads:DNA ratio, according to manufacturer's instructions. DNA was eluted in water (average 10 Kb length as per 1% agarose gel/TAE analysis). The DNA was end-repaired and A-tailed with Ultra II End Prep kit (New England Biolabs) and DNA-sequencing library prepared using SQK-LSK114, as previously described[52], and sequenced using a MinION. A total of 2,088,660 sequences with a N50 of 1.54 Kb and mean Q-score quality of 11.

## Hi-C and Pore-C computational analysis

Raw fastq files were mapped to the *T. brucei* 427 genome[7] (data shown for haplotype A) using bwa-mem version 0.7.18[61] using Hi-C mapping parameters bwa mem -5SPM. DNA contacts were obtained using Pair-Tools version 1.1.3[41] with parse --walks-policy all parameters and filtering uniquely mapped reads and removal of duplicates. Pair files were converted to cool files using cooler version 0.10.2[62], using cooler cload pairs parameters. For Pore-C, fastq sequences were aligned to the genome using minimap2 version 2.28[63]. DNA contacts were obtained using PairTools, using parse2 command, −single-end para-meter, and filtering uniquely mapped reads and removal of duplicates. Pair files were converted to cool files using cooler cload. The matrices of three biological replicates were combined and corrected using HiCExplorer version 3.7.6[64] (detailed below). Using HiCExplorer hic-SumMatrices tool, matrices were binned at 10 Kb and 50 Kb resolution. Matrices were balanced using the Knight-Ruiz (KR) method, as implemented in hicCorrectMatrix. For comparison, we also corrected matrices using iterative correction and eigenvector decomposition (ICE) with hicCorrectMatrix, using filterThreshold values of −1.5 and 5. We did not observe significant differences between KR and ICE bal-ancing upon matrix visual and content inspection. Hence, further analyses were obtained using the KR balancing method. To compare different datasets, matrices were normalized to the condition with the smallest matrix using the hicNormalize tool. To compare, we also normalized matrices from 0 to 1 and did not detect differences between the two normalization results. The results shown were obtained by normalizing to the smallest matrix. Compartments were calculated using FAN-C version 0.9.28[35]. Briefly, we used the fanc compartments command to identify A and B compartments from a 50 kb resolution Hi-C matrix, where A has positive values and B has negative values. Compartments were uniformized based on the gen-ome's GC content (option -g), with A having high GC content and B having low GC content. The eigenvector values were then written to the file (option -v). Then, using RNA-seq of *T. brucei* bloodstream forms[14], we associated transcribed and silent regions according to A and B compartments, respectively. TADs and loops were analyzed with HiCExplorer[65]. TADs were searched using hicFindTADs using a 1Kb, 10 Kb, and 50Kb resolution matrix, a delta of 0.01, threshold of 0.05, multiple test correction set to false discovery rate, minimum and maximum depth of 3 times matrix resolution and 10 times resolution, respectively, and steps equal to matrix resolution, using a KR balanced matrix. Afterwards, identified TADs were merged using hicMergeDo-mains. We also analyzed TADs and loops using an ICE balanced matrix, with the same parameters as above, without significant differences between ICE and KR, although KR balancing resulted in more reliable detection of TADs after their inspection and comparison among datasets. The results shown in the manuscript and all subsequent

analyses were performed using a KR balanced matrix. To compare inter- and intra-TADs, we used the hicInterIntraTAD tool with the TAD domains identified by hicFindTADs (as specified above). A comparison between tet+ and tet− was performed using HiCExplorer, utilizing the KR-balanced and normalized matrices (as described above) at a 10 Kb resolution. For loops, we used hicDetectLoops, using a 10 Kb matrix, maximum loop distance of 2 Mb, $p$ value of 0.025, peak region width of 2 bins, and a $p$ value pre-selection threshold of 0.1. We used hicCompareMatrices to compare the differences in contact between two matrices, using the log2 ratio of the normalized matrices. To quantify interactions among compartments, we performed analyses with Pentad[66]. We used the get_pentad_cis.py script with a KR-balanced matrix and a cutoff distance of 100,000 kb, i.e., short-range contacts are lower than the cutoff and long-range contacts are higher than the cutoff. The A and B compartments, as explained above, were used to define compartments. For trans-contacts, we used get_pentad_trans.py. Compartment strengths for cis and trans-contacts were analyzed using quant_strength_cis.py or quant_strength_trans.py, respectively. To analyze contact between compartment boundaries, boundary sequences were selected based on 10–100 Kb sequences between core (A) and subtelomeric (B) compartments. Boundary contacts were searched from a KR balanced matrix and normalized to the smallest dataset. RAP1 ChIP-seq data were used to verify boundary regions. Matrix contacts were extracted into.txt format using cooler dump, and contact counts were obtained using scripts find_contacts_sum.py to extract contact counts, and run_contact_stats.py to obtain contact statistics (total count, mean count, standard deviation, median) for specific boundaries and outside boundaries (background). Scripts are available at https://github.com/cestari-lab/lab_scripts.

### ChIP-seq and RNA-seq computational analysis

ChIP-seq and RNA-seq data were obtained from refs. 14,30,33, and analyzed as previously described in ref. 14. All scripts are available at https://github.com/cestari-lab/lab_scripts. The RNA-seq or ChIP-seq fastq sequences were mapped to *T. brucei* 427–2018 reference genome (haplotype A) using minimap2 version 2.28. For nanopore data, reads with a Q-score >7 were used for analysis, and the mean Q-score was 12. Alignments were processed from SAM to BAM files with SAMtools version 1.21. Supplementary and secondary alignments were filtered out using samtools flags (-F 2304). RNA-seq mapped reads were quantified using featureCounts from the package Subread and used for differential expression analysis using EdgeR version 4.8.0[67]. For ChIP-seq, aligned reads were mapped with minimap2 and processed with deepTools[68] for coverage analysis using bamCoverage and enrichment analysis using bamCompare comparing RAP1-HA ChIP vs Input. Peak calling and statistical analysis were performed for broad peaks with Model-based Analysis of ChIP-Seq MACS3 version 3.0.3[69], and data were visualized using the Integrated Genomics Viewer tool version 2.17.4 (IGV, Broad Institute). To determine if RAP1-HA mapping to boundaries resulted from multiple mapping reads, we selected primary alignments, as well as those with mapping probabilities of 90% (mapQ 10). To identify RAP1 binding sequences, boundary peaks were identified in IGV, and the corresponding sequences were retrieved from the genome using the script getdna.py. To identify motifs in boundary sequences, we analyzed sequences using Multiple Em for Motif Elicitation (MEME) tool version 5.5.8[70] using default parameters in the classic mode, except for site distribution set to any number of repetitions (anr) and number of motifs set to 100.

### Surface plasmon resonance

Ten µg/ml of RAP1-His (110 nM) were diluted in binding buffer (20 mM HEPES [(4-(2-hydroxyethyl)-1-piperazineethanesulfonic acid)], 150 mM NaCl, 10 mM KCl, 10 mM MgCl$_2$, 10 mM CaCl$_2$ and 0.2% NP-40) for binding assays. The NTA-sensor (Nicoya) surface was cleaned in a solution of 10 mM HCl and 350 mM EDTA solution, then activated in 40 mM NiCl$_2$ solution. RAP1-His was immobilized in the NTA sensor in binding buffer, followed by sensor blocking in binding buffer supplemented with 0.5% BSA. PIP5Pase-V5 diluted in binding buffer was added in various concentrations (0.0008 nM, 0.004, 0.008, 0.016 nM) for binding kinetic analysis. The sensor surface was regenerated for every binding assay using 500 mM Imidazole diluted in Nanopure MilliQ water. Reactions were prepared in 200 µL volume in 96-well plate and run in the OpenSPR-XT (Nicoya). Data was recorded and analyzed using the Traceviewer software version 4.5.8846.19906 (Nicoya).

### Statistics and reproducibility

Hi-C, ChIP-seq, and RNA-seq were performed in three biological replicates. In vivo XLMS and PIP5Pase-V5 immunoprecipitations were performed 11 times. RAP1-HA immunofluorescence and SPR were performed three times. SDS/PAGE and Western blots are presented as representative experiments with similar results obtained from at least three replicates. Sample sizes are indicated in the figure legends. Comparison between groups was performed using a two-sample two-sided t-test with $\alpha = 0.05$ and a 95% confidence level, determined by statistical and graphical software OriginPro 2024b, unless otherwise stated. Exact $p$-values are indicated in the figures, unless otherwise stated. Box plots were generated using OriginPro version 2024b. Protein interaction networks were created using Cytoscape version 3.9.1, and the plugin XlinkCyNET version 1.5.3 was utilized for XLMS interaction visualization. Hi-C, RNA-seq, and ChIP-seq graphs were obtained using HiCExplorer version 3.7.6 or pyGenomeTracks version 3.9.

### Reporting summary

Further information on research design is available in the Nature Portfolio Reporting Summary linked to this article.

## Data availability

RNA-seq and ChIP-seq sequencing data are available in the Sequence Read Archive (SRA) with the BioProject identification PRJNA934938 [https://www.ncbi.nlm.nih.gov/bioproject/934938]. Hi-C sequencing data is available in the SRA with BioProject identification PRJNA1198910. The mass spectrometry proteomics data have been deposited in the ProteomeXchange Consortium via the PRIDE partner repository, with the dataset identifier PXD059635. Source data is also available with this paper. All unique biological materials, such as genetic cell lines and plasmids, are available upon request. Source data are provided with this paper.

## Code availability

Codes used for data analysis are available at https://github.com/cestari-lab/[71].

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

## Acknowledgements

Canadian Institutes of Health Research grant CIHR PJT-175222 (IC). The Natural Sciences and Engineering Research Council of Canada grant RGPIN-2019-05271 (IC). Fonds de Recherche du Québec - Nature et technologie grant 2021-NC-288072 (IC). Canada Foundation for Innovation grant JELF 258389 (IC). FRQNT-Ukraine postdoctoral fellowship BUKX:2022-2023 337989 (OK). The Natural Sciences and Engineering Research Council of Canada CGS M fellowship (LBA). FRQNT doctoral training scholarship 2024-2025-B2X-345472 (TI). This research was partly enabled by computational resources provided by Calcul Quebec (https://www.calculquebec.ca/en/) and the Digital Research Alliance of Canada (alliancecan.ca). We thank Dr. Suzanne McDermott (Center for Global Infectious Disease Research, Seattle Children's) for reading the manuscript and providing valuable suggestions.

## Author contributions

T.I. performed cross-link and mass spectrometry, SPR protein binding assays, and microscopy experiments; L.B.A. optimized Hi-C protocol, performed Hi-C and Pore-C, designed Hi-C and Pore-C computational pipeline, analyzed Pore-C dataset, and performed ChIP-seq analysis; O.K. optimized pore-C protocol and preliminary data analysis; I.C. designed research, generated cell lines, performed Hi-C, Pore-C, and ChIP-seq computational analysis. I.C. obtained research funds, supervised the research, and wrote the manuscript. All authors read and revised the manuscript.

## Competing interests

The authors declare no competing interests.
