## [Transparent Peer review file · Nature Communications]

Chromosome compartment assembly is essential for subtelomeric gene silencing in trypanosomes

Corresponding Author: Professor Igor Cestari

Version 0:

Reviewer comments:

Reviewer #1

(Remarks to the Author)

The manuscript by Antunes et al. is highly interesting and presents novel and valuable insights that have the potential to significantly advance our understanding of chromatin architecture, 3D genome organization in *T. brucei*, and its role in VSG regulation. The study combines complementary methodologies and provides a rich dataset that will certainly contribute to the field. The manuscript is generally well written, clear, however, the lack of additional methodological details and clarifications significantly limits a thorough evaluation of the work. While several findings are undoubtedly novel, others appear to rely on the re-analysis or reinterpretation of previously published data. For a non-expert in VSG regulation, some of these reinterpretations seem largely equivalent to prior knowledge or represent more of a semantic refinement rather than a conceptual advance. For instance, the ChIP-seq of RAP1 had already been performed by the authors in previous studies, demonstrating its enrichment at subtelomeric regions. Given that subtelomeric regions form a distinct compartment from the core genome (as shown by Müller et al.), the observation that RAP1 is enriched within this compartment is somewhat expected. Likewise, the interaction between PIP5Pase and RAP1 has already been demonstrated in previous work.

1. I understand that the authors have been working with on RPA1 and PIP5Pase for a long time, generating different cell lines (many used in this manuscript); however, the initial argument for using cell lines expressing PIP5ase-tagged for detection of a “spatial chromatin protein network” (first subsection of the Results) is not fully convincing. If the aim is to describe a chromatin spatial network or to identify chromatin regulatory proteins (as stated), it would be more appropriate to focus on classical chromatin components, such as histones or well-established chromatin remodelers. I strongly recommend that the authors rewrite or reorganize this section to clarify the rationale and strengthen the argument.

2. XL-MS Experiments:

While I understand that the XL-MS assay detects protein-protein interactions and that the experiment was based on an IP targeting PIP5Pase (thus expected to capture its interactors preferentially), it is noteworthy that very few cross-links are associated with PIP5Pase itself (only four proteins, according to Table S2). In contrast, RAP1 — which was not the bait for the IP — shows 14 cross-linked proteins.

Despite the fact that previous studies, including the SPR data shown in this manuscript, clearly demonstrate interactions between RAP1 and PIP5Pase, the XL-MS approach failed to detect this interaction. Furthermore, the total number of detected proteins (494) seems unusually high, raising concerns about data quality and the stringency of the experimental setup. This brings up an important question: were proper controls included to reduce false positives? Specifically: Were negative controls used, such as samples without crosslinker (DSS) or non-immunoprecipitated samples? Including such controls could significantly reduce background and improve data robustness and specificity.

Additionally, I had difficulty tracing the numbers reported in the manuscript back to the corresponding tables. For example: Line 116: “PIP5Pase direct and counterpart cross-links revealed a subnetwork of 204 proteins with 750 cross-links (Fig. 1F, Table S1 and S2)”. I could not find the list of 204 proteins nor the 25 proteins with multiple cross-links in Table S1 or S2 as indicated.

Important information is also missing regarding the MS analysis parameters:

What were the fixed and variable modifications used to account for the mass shifts caused by crosslinking?

How much protein (in micrograms) was used for digestion? The authors mention starting with 2×10^{10} cells — how much total protein is recovered after extraction and IP?

These details are essential for reproducibility and for other labs aiming to incorporate XL-MS into their workflow.

Another point requiring clarification: The methods mention MS analysis for procyclic forms, but the Results focus on bloodstream forms (BSF). Were the datasets combined? Please specify in the figure legends which life stage each dataset refers to and justify why different conditions were used for each form.

Figures and Tables (XL-MS):

Fig. 1E – 918 XL: What does this number represent? Number of cross-links? Proteins?

Fig. S1A: Why does the number of cross-linked proteins increase progressively and then drop sharply at experiment number 12? Please clarify.

3. Hi-C and Pore-C Data:

Table S4: Does it represent the sum of replicates? This should be clarified in the caption.

Fig. 3C: How were the A and B compartments defined? The authors mention using FANC software, but no details are provided regarding the criteria or thresholds. This must be described in the Methods section.

Fig. 3D: How were short-range and long-range contacts defined? Please specify the distance cutoffs used.

Which version of the *T. brucei* genome (FASTA file) was used?

What Illumina sequencer and read length were used for Hi-C?

Regarding Pore-C, there is no description of the protocol in the Methods section. Additionally, the text states that “DNA fragments ranging from 100–450 bp were size-selected”, which seems inconsistent with the expected advantages of Pore-C, typically designed to capture much longer DNA fragments. At line 186, the authors report an N50 of 1.54 kb — please clarify how this was achieved and discuss whether such short fragments truly justify the use of Pore-C over conventional Hi-C.

4. Concerns About Definitions and Interpretation:

The distinction between “compartment” referring to A/B compartments (Hi-C based) and core/subtelomeric regions (genomic context) is unclear throughout the manuscript. For example, in Fig. 3G, the authors refer to two compartments associated with transcribed and silent regions. Are these the same as the A/B compartments determined by FANC?

Has it already been demonstrated that A and B compartments correspond to core and subtelomeric regions in *T. brucei*? If so, please cite the relevant studies. If not, the authors should explicitly show whether or not such correspondence exists.

How were sub-TADs defined? Also long and short interactions? What criteria were used?

Fig. 4G appears to be missing.

Fig. 4E is referenced before Fig. 4A-C; please check the figure order. Also, confirm whether the color descriptions (red vs. green) are correct.

In Fig. 3G (inset): It is well established that many chromatin-associated proteins localize to divergent strand switch regions (dSSRs) in *T. brucei*. The authors should discuss whether the observed enrichment at boundaries reflects an association with dSSRs. It would be informative to annotate PTU boundaries directly on the heatmaps shown (e.g., Fig. 3G).

Interestingly, proteins like BDF2, HDAC, HAT1, and ZCW1 flank RAP1 at boundaries. This is a noteworthy observation that deserves further discussion. Does this pattern occur at multiple boundaries? Are these proteins consistently located at TAD boundaries?

5. Repetitive Regions and Potential Artifacts:

The authors should investigate whether boundaries are enriched in repetitive elements. Could some of the observed boundary-to-boundary interactions be artifacts resulting from the presence of highly repetitive subtelomeric regions?

It would strengthen the manuscript if the authors analyzed sequence similarity between boundaries, searched for motifs, and discussed how they handled multimapping reads during Hi-C and Pore-C data processing.

6. Data Normalization:

Tables S3 and S4: The number of Hi-C contacts varies significantly between replicates (e.g., PIP5ase Tet⁺ ranges from ~70K to ~116K contacts), seemingly independent of the number of reads. Is this expected? Please explain.

Related to this, before calculating log₂FC values (Fig. 5B), shouldn't the authors normalize for the same number of reads between Tet⁺ and Tet⁻ samples? There are several established metrics for comparing Hi-C matrices that account for varying sequencing depths — please clarify which normalization strategy was used.

(Remarks on code availability)

Reviewer #2

(Remarks to the Author)

This study provides significant contributions to the scientific understanding of chromosome organization and gene silencing in *Trypanosoma brucei*. It reveals that subtelomeric gene silencing—a mechanism essential for antigenic variation and immune evasion—relies on the spatial segregation of chromatin into distinct chromosome compartments. By identifying and characterizing key chromatin-associated proteins such as RAP1, the work demonstrates that these components regulate the organization of repressed and active regions of the genome. Moreover, it establishes a novel connection between phosphoinositide signaling and chromatin structure, suggesting that lipid-mediated regulation may play a broader role in nuclear architecture. These findings provide new insights into gene regulation mechanisms in eukaryotic parasites and potentially other organisms, representing a significant step toward understanding epigenetic control in complex pathogens. Although this work advances our knowledge of chromatin compartmentalization, we have questions and comments we detail below:

- The authors must specify how the Hi-C data matrices were normalized. The algorithm/method used in the Methods section must be detailed, as normalization is a critical step in the analysis that significantly impacts the results.

- The heatmaps presented contain several regions with limited contact information. Which resolution was used to analyze domain formation and loop structures? The authors should indicate the resolution of each heatmap, either within the figures or in the corresponding legends, as well as specify the resolution used for each type of analysis.

- Are the contact frequencies in figures 3B, 3C, 3G, 5A, 5B, 5C, 5G, and 6A normalized? If so, please indicate them in the figures.

-It would be interesting to include additional examples similar to Figure 3C to illustrate how compartments are organized across other chromosomes. The authors are encouraged to provide more examples as supplementary figures.

- The authors should include a supplementary table listing the predicted domains and loops to ensure this information is accessible to the scientific community.

-The authors should present additional genomic regions to better assess the distribution of chromatin-associated proteins, rather than limiting the analysis to a single locus as shown for BDF2, HAT1, HDAC1, and ZCW1. Furthermore, it would be helpful to visualize RAP1 distribution across other chromosomes. Please include supplementary figures with additional examples to support a more comprehensive evaluation.

-Figures 3e and 5e require a more detailed explanation. The authors should clarify how these figures were generated, including the methods used for their calculation, and clearly describe what the data represent.

-Line 240, correct 4G for 5G.

- Although the authors have made their scripts available on GitHub, they should also specify the algorithms used and provide a general description of the RNA-seq and ChIP-seq analysis methods within the manuscript.

- The RNA-seq results should be better presented, as the current figures do not allow for clear visualization of the +/- Tet effect on global gene expression within the compartments, as described in the text.

- The formation of chromatin domains in both *T. cruzi* and *T. brucei* has been previously described (DOI: 10.1038/s41564-023-01483-y), including a comparative analysis of domain organization across different life stages of *T. brucei*. The authors should incorporate this background into the Introduction or Discussion to provide better context for the current findings.

- The Hi-C data is not available at the accession code provided in the manuscript.

- It is important for the scientific community to establish a unified nomenclature for chromatin domains in trypanosomes. *T. cruzi* and *T. brucei* previously described these domains as Chromatin Folding Domains (CFDs). The authors should clarify their choice to refer to them as TADs and discuss how this terminology relates to or differs from previously proposed definitions.

- Throughout the manuscript, the authors refer to PTUs (polycistronic transcription units), when in our view they should instead refer to DGC (directional gene clusters). Although the two concepts are related, they are not interchangeable: PTUs are defined by their transcriptional activity, whereas DGCs refer to blocks of coding genes oriented in the same direction, regardless of whether they are actively transcribed.

(Remarks on code availability)

Reviewer #3

(Remarks to the Author)

(Remarks on code availability)

Version 1:

Reviewer comments:

Reviewer #1

(Remarks to the Author)

The authors have satisfactorily addressed most of the questions. However, as I mentioned before, although the manuscript indeed contains new data that advances the field, it is interspersed with results already shown either by the authors themselves or by others. Additionally, the compartments findings has a critical semantic issue that must be addressed to avoid confusion. Thus, it is up to the editor to evaluate whether this constitutes a justified novelty to be published in this journal.

The major semantic issue of this manuscript concerns the previous definition of compartments (subtelomeric/telomeric vs. core compartments), and now the classical definition of A and B compartments from the Hi-C topological organization; and

the fact that both compartment definition is indeed referring to the same class of genes/location in the genome. As it is currently written, parts of the text may give the impression that the authors are simply reiterating what has already been shown by Nicolai Siegel's group (Müller et al.; Rabuffo et al., etc.). However, I understand that the authors are actually referring to the A and B compartments identified using the standard Hi-C bioinformatics pipeline. To avoid confusion, I strongly recommend that this compartment definition distinction be clearly stated at the beginning of the manuscript.

For example, in Muller et al. (paper from Nicolai Siegel's lab) they showed that "Genome-wide chromosome conformation capture (Hi-C) reveals a distinct partitioning of the genome, with antigen-encoding subtelomeric regions that are folded into distinct, highly compact compartments." Thus, the existence of core compartments (highly transcribed, fewer DNA-DNA contacts) and antigen-encoding subtelomeric compartments (silent, more DNA-DNA contacts) was already established. I understand that the main advance here is that their Hi-C and Pore-C assays now reveal the presence of A and B compartments, TADs and loops (although Viraque et al. 2024 also showed part of this) by applying standard Hi-C computational pipelines. Therefore, statements such as "- We show that *T. brucei* chromosomes are organized into core transcribed and silent subtelomeric compartments,..." (abstract - line 17), or at lines 198–199 "Thus, we performed Hi-C and ChIP-seq analysis in *T. brucei* bloodstream forms. *T. brucei* chromosomes are organized into core transcribed and subtelomeric repressed regions, with some chromosomes containing ESs (Fig. 3A)." do not seem appropriate without citing the original work. In my view, the novelty is that they confirm the presence of compartments already reported and now refine this observation using proper bioinformatic analysis of Hi-C data (such as FANC).

Another example: At line 208 they write: "Notably, in *T. brucei*, the organization of A and B compartments correlates well with core and subtelomeric regions for Mb-long chromosomes." A more precise statement would be something like: "Notably, in *T. brucei*, the organization of A and B compartments correlates well with core and subtelomeric regions for Mb-long chromosomes – 3D compartments already described before (Muller et al.)."

Another important point:

In their reply, the authors state: "The binding of RAP1 became evident to us only when analyzing RAP1 ChIP-seq using a genome version in which subtelomeric regions were co-assembled with core regions." However, it has already been shown that RAP1 binds (and unbinds) to subtelomeric regions in TbPIP5Pase mutants (see Fig. 5 and its legend of Touray et al. 2023).

Finally, the information that perturbation of inositol metabolism affects the expression of silent VSGs was already published by this group: "Conditional knockdown of phosphatidylinositol 5-kinase (TbPIP5K) or phosphatidylinositol 5-phosphatase (TbPIP5Pase) or overexpression of phospholipase C (TbPLC) derepresses numerous silent ESs in *T. brucei* bloodstream forms." (doi: 10.1073/pnas.1501206112). The way it is written, makes it sound like a novel finding (see abstract, lines 22–23: "Inactivation of the RAP1 regulator, phosphatidylinositol phosphate 5-phosphatase, removes RAP1 from boundaries and subtelomeric compartments, disrupting chromatin compartment contacts and activating all VSG genes."). The truly new finding here is that they show that inactivation of phosphatidylinositol phosphate 5-phosphatase removes RAP1 from boundaries and subtelomeric compartments, disrupting chromatin compartment contacts. The activation of VSG expression, however, had already been reported by the group.

Minor points:

What does the circle composed of several small green circles in Fig. S2C represent?

In the virtual 4C analysis of chromosomes 11 and 10 (Fig. 4C and 4), the figure could be improved, perhaps by including a scheme showing where the chromosome 10 and 11 boundaries lie on the x-axis.

In Fig. 5C, is there a normalization issue? The Tet– Hi-C matrix clearly shows no compartments, yet the plots suggest boundaries. Note also that the scales of Tet+ and Tet– are not the same, which makes the comparison even more difficult.

What is the phenotype of Tet– cells? (viability? infectivity?)

Lines 301–302: "The data indicate that PIP5Pase also coordinates subtelomeric compartment interactions, which may occur via the association..." → this should read "the data SUGGEST that PIP5Pase..." since the authors have not shown by orthogonal methods that proteins of the PIP5Pase network (except for BDF2, HAT, etc., which also bind elsewhere in the genome) indeed bind to compartment boundaries (e.g., by ChIP-seq or KO).

Lines 321–323 seem contradictory:

"The data suggest that contact among boundaries is not necessary to repress silent subtelomeric regions, but it may help demarcate core and subtelomeric compartments. The data indicate that the assembly of chromosome compartments is essential for silencing subtelomeric VSG genes."

Fig 6^a : RNA-seq ratio between wt/mutant seem opposite

(Remarks on code availability)

Reviewer #2

(Remarks to the Author)

The authors have sufficiently addressed the concerns raised in the previous round of review and have incorporated the missing information into the manuscript. However, there remain a few additional comments that we would like to put forward for consideration.

Authors:

"It is important for the scientific community to establish a unified nomenclature for chromatin domains in trypanosomes. *T. cruzi* and *T. brucei* previously described these domains as Chromatin Folding Domains (CFDs). The authors should clarify their choice to refer to them as TADs and discuss how this terminology relates to or differs from previously proposed definitions. Author: The terminology TAD has been used in all fields of biology (human, yeast, *C. elegans*, *Drosophila*, etc) to describe a segment of genomic DNA sequence co-interacting more frequently than with sequences outside the specific segment as obtained from Hi-C experiments (Szabo et al. 2019, Science Advances). The computational algorithms used in this work were designed to identify TADs using Hi-C data, and thus, we report them as found. The term CFDs has only been used in a single paper (Díaz-Viraqué et al. 2023, Nat Microb). The authors used algorithms designed to identify TADs (i.e., TADtools) in Hi-C data, but renamed the identified TADs to CFDs based on their length (~129 Kb). The term CFD refers to TADs identified with algorithms and methodology established for TADs in the broader field of biology (Dixon et al. 2012, Nature; Beagan and Phillips-Cremins, 2020, Nat Genetics; Szabo et al. 2019, Science Advances). For example, TADs of ~200 Kb were identified in yeast (Eser et al. 2017, PNAS), and TADs of 135 to 215 Kb in *Drosophila* species (Torosin et al. 2022, Mol Biol Evol), which are similar in size to those identified in *T. cruzi*. Díaz-Viraqué et al. paper (Fig. 5B) shows *T. brucei* TADs ranging 150-400 Kb, which is consistent with what we report and found by others (Dixon et al. 2012, Nature; Rowley and Corces, 2018, Nat Rev Genet). Hence, TADs seem to have a typical range of 100-1000 Kb in various organisms. We opted for the TAD terminology because it is consistent with the methods (Hi-C data and algorithms used) and aligns with the term usage in the broader field."

ANSWER:

The use of algorithms designed for TAD identification does not in itself justify calling the domains observed in trypanosomes TADs. The conceptualization of chromatin structures in microorganisms has evolved differently. Due to differences in genome size and biological features (such as the absence of one promoter per gene and enhancer-promoter regulation), the structural domains identified in trypanosomes were referred to as chromatin folding domains (CFDs). The genomic architecture of these parasites differs substantially from that of multicellular eukaryotes, lacking promoter-enhancer regulation, introns, and large intergenic regions. Accordingly, the domains identified by Hi-C in *T. cruzi* have been described as chromatin folding domains (CFDs), ranging from 20 to 940 kb (average ~129 kb), a size comparable to chromatin interaction domains (CIDs) in bacteria (30–420 kb, average ~170 kb; Dekker et al., 2015; Le et al., 2013). These values differ from those reported for TADs in multicellular eukaryotes (average 880 kb; Dixon et al., 2012), most likely reflecting the reduced extent of non-coding regions in trypanosomatids (including the absence of introns and RNA polymerase II regulatory elements, as well as shorter intergenic and inter-DGC regions). The manuscript would be strengthened if the authors explicitly clarified how their use of the term "TAD" relates to the previously

AUTHORS:

"The failure of other studies in identifying TADs in *T. brucei* may reflect a combination of Hi-C sequencing depth and, thus, the obtained number of contacts and the approach used. Although analysis of the same dataset by others reported the presence of such domains, also found in *T. cruzi*."

ANSWER

The original sentence is unclear and needs revision for clarity. The use of "Although" makes the statement vague, especially considering previous comments on chromatin folding domains (CFDs). Reference 45 not only shows CFDs in *Cruzi* but also indicates the presence of CFDs in *T. brucei*, with sizes and magnitudes similar to those reported in *T. cruzi*. Additionally, that work demonstrates conservation of CFD in procyclic and bloodstream forms and a correlation with global transcription, suggesting a common feature of trypanosomatids. Like in *T. cruzi*, there is no exact correlation between DGC and PTU. Therefore, there was no actual 'failure' to identify these domains in *T. brucei*. This point should be included in the discussion rather than presenting the results as a new finding.

(Remarks on code availability)

We have already done it in the previous round.

Reviewer #3

(Remarks to the Author)

(Remarks on code availability)

Version 2:

Reviewer comments:

Reviewer #1

(Remarks to the Author)

The authors have addressed all issues satisfactorily.

(Remarks on code availability)

Reviewer #2

(Remarks to the Author)

In my opinion, the manuscript is now suitable for acceptance.

(Remarks on code availability)

Reviewer #3

(Remarks to the Author)

(Remarks on code availability)

Response to reviewer's comments. Reviewer's comments in black, author's responses in blue.

Reviewer #1 (Remarks to the Author):

The manuscript by Antunes et al. is highly interesting and presents novel and valuable insights that have the potential to significantly advance our understanding of chromatin architecture, 3D genome organization in *T. brucei*, and its role in VSG regulation. The study combines complementary methodologies and provides a rich dataset that will certainly contribute to the field. The manuscript is generally well written, clear, however, the lack of additional methodological details and clarifications significantly limits a thorough evaluation of the work. While several findings are undoubtedly novel, others appear to rely on the re-analysis or reinterpretation of previously published data. For a non-expert in VSG regulation, some of these reinterpretations seem largely equivalent to prior knowledge or represent more of a semantic refinement rather than a conceptual advance. For instance, the ChIP-seq of RAP1 had already been performed by the authors in previous studies, demonstrating its enrichment at subtelomeric regions. Given that subtelomeric regions form a distinct compartment from the core genome (as shown by Müller et al.), the observation that RAP1 is enriched within this compartment is somewhat expected. Likewise, the interaction between PIP5Pase and RAP1 has already been demonstrated in previous work.

Author: We appreciate the reviewer's comments and thorough evaluation of our work. To clarify, this work demonstrates that RAP1 binds to chromosome compartment boundaries and spreads over subtelomeric regions. Our previous work (Touray et al. 2023, eLife) focused on telomeric expression sites. Moreover, we did not have the Hi-C data, and thus we were unaware of RAP1 binding to the compartment boundaries (we were not aware of the boundaries). The binding of RAP1 became evident to us only when analyzing RAP1 ChIP-seq using a genome version in which subtelomeric regions were co-assembled with core regions. Then, we noticed the contrast between RAP1 binding to subtelomeric vs core chromosome regions and enrichment at the compartment boundaries. Moreover, we demonstrate here that RAP1 represses subtelomeric VSG genes. This results from PIP5Pase knockdown, which leads to RAP1 removal from boundaries and subtelomeric regions. Although we knew that PIP5Pase knockdown affected transcription of subtelomeric VSG genes, we did not know the mechanism underlying the silencing of subtelomeric VSG genes. The results presented here are novel and have not been reported previously by us or others, and they include a mechanistic understanding of how subtelomeric VSG genes are silenced.

1. I understand that the authors have been working with on RPA1 and PIP5Pase for a long time, generating different cell lines (many used in this manuscript); however, the initial argument for using cell lines expressing PIP5ase-tagged for detection of a "spatial chromatin protein network" (first subsection of the Results) is not fully convincing. If the aim is to describe a chromatin spatial network or to identify chromatin regulatory proteins (as stated), it would be more appropriate to focus on classical chromatin components, such as histones or well-established chromatin remodelers. I strongly recommend that the authors rewrite or reorganize this section to clarify the rationale and strengthen the argument.

Author: We appreciate the reviewer's recommendation. We have established a role for PIP5Pase in regulating chromatin organization (this work, and Cestari and Stuart, 2015, PNAS; Cestari et al., 2019, Mol Cell Bio.; Touray et al. 2023, eLife). Its regulatory function entails controlling levels of PI(3,4,5)P3, which bind to RAP1 and regulate its DNA binding (Touray et al. 2023, eLife) rather than direct modification of chromatin. Our goal was to identify proteins associated with PIP5Pase to identify additional proteins with signalling or regulatory roles in chromatin organization. In agreement with the reviewer's point, we reworded the heading as (line 94) "**Signalling and chromatin regulatory cross-link network at ~30 Å resolution**", which focuses on proteins with signalling and regulatory roles rather than chromatin modification. We also added some text at the beginning of the session to clarify the goal of this session, lines 95-97: "*We performed in vivo chemical cross-linking and immunoprecipitation of PIP5Pase followed by mass spectrometry (XLMS) to identify additional proteins potentially involved in signalling and chromatin regulation.*".

2. XL-MS Experiments:

While I understand that the XL-MS assay detects protein-protein interactions and that the experiment was based on an IP targeting PIP5Pase (thus expected to capture its interactors preferentially), it is noteworthy that very few cross-links are associated with PIP5Pase itself (only four proteins, according to Table S2). In contrast, RAP1 — which was not the bait for the IP — shows 14 cross-linked proteins. Despite the fact that previous studies, including the SPR data shown in this manuscript, clearly demonstrate interactions between RAP1 and PIP5Pase, #1) the XL-MS approach failed to detect this interaction. Furthermore, the total number of detected proteins (494) seems unusually high, raising concerns about data quality and the stringency of the experimental setup. This brings up an important question: were proper controls included to reduce false positives? Specifically: Were negative controls used, such as samples without crosslinker (DSS) or non-immunoprecipitated samples? Including such controls could significantly reduce background and improve data robustness

and specificity. #2) Additionally, I had difficulty tracing the numbers reported in the manuscript back to the corresponding tables. For example: Line 116: “PIP5Pase direct and counterpart cross-links revealed a subnetwork of 204 proteins with 750 cross-links (Fig. 1F, Table S1 and S2)”. I could not find the list of 204 proteins nor the 25 proteins with multiple cross-links in Table S1 or S2 as indicated. #3) Important information is also missing regarding the MS analysis parameters: What were the fixed and variable modifications used to account for the mass shifts caused by crosslinking? How much protein (in micrograms) was used for digestion? The authors mention starting with 2×10^{10} cells — how much total protein is recovered after extraction and IP? These details are essential for reproducibility and for other labs aiming to incorporate XL-MS into their workflow. #4) Another point requiring clarification: The methods mention MS analysis for procyclic forms, but the Results focus on bloodstream forms (BSF). Were the datasets combined? Please specify in the figure legends which life stage each dataset refers to and justify why different conditions were used for each form.

Author: We numbered the questions to address all points. #1: The XLMS is a proximity-based approach, and it requires DSS cross-linkable amino acids between two proteins within a 20-30 Å distance, which makes the approach very stringent. This is an uncontrollable variable and a limitation of the approach. If interactions between two proteins are transient, the chances of detecting cross-links also decrease.

On PIP5Pase cross-links: Analysis of XLMS showed a total of 373 peptides for PIP5Pase, six of which were cross-linked. In contrast, we found 241 peptides for RAP1, of which 39 were cross-linked (Supplementary Table 1). On average, we obtained 13 ± 14.1 cross-links per protein, with some proteins having up to ~ 100 cross-links (Supplementary Fig. 1), and a total of 66,820 cross-linked peptides with a frequency of 12.41 ± 14.68 (min = 1, max = 388) protein detection (new Supplementary Fig. 1). The data indicate that PIP5Pase is poorly cross-linked by DSS. We also performed an *in vitro* XLMS after PIP5Pase immunoprecipitation. We identified 57 peptides for PIP5Pase, but no additional cross-links were observed (new Supplementary Table 5). We have another study (not included here) in which we performed total cell XLMS in *T. brucei* bloodstream forms with DSS (8 replicates) and EGS (6 replicates). We found 1.1 million cross-linked peptides and 67,302 protein cross-links. We found many peptides for PIP5Pase, but none of them were cross-linked; however, we identified cross-links for RAP1 and other proteins. We argue that this is an intrinsic characteristic of this protein, perhaps because of its folding limiting DSS cross-linkable amino acids on its surface.

On PIP5Pase enrichment and controls: We included new data to calculate PIP5Pase enrichment over a background control. We found that, despite low cross-links, the protein was enriched, which makes its interactions still relevant. We show this by Western blot (Fig. 1C) and enrichment analysis of the mass spectrometry (new Supplementary Fig. S2 and Supplementary Table 2). In this analysis, we compared total identified peptides from XLMS IPs from cells expressing PIP5Pase-V5 with cells without V5-tagged PIP5Pase (as a control, 4 biological replicates). The enrichment analysis demonstrates PIP5Pase was enriched (\log_2 fold-change = 3.66, p-value = 0.00029). In a previous work (Cestari *et al.*, 2019, *Mol Cell Biol*), we performed extensive PIP5Pase and RAP1 IPMS (without cross-linking). Analysis of the top 30 proteins co-immunoprecipitated was also found in this work using XLMS (Supplementary Fig. 2), indicating an overlap between both datasets and implying that the current dataset is not an artifact. Moreover, Staneva *et al.* 2021, *Genome Research* (see this paper's Supplementary Tables) showed interaction by IPMS of PIP5Pase, RAP1, HDAC1, HDAC3, BDF2, HAT1, TRF, H2A, SMC, RCF, SNF2, RepC, among other chromatin-associated proteins. These proteins were also identified in our work by XLMS (Fig. 2C), showing a remarkable overlap in the datasets. This further supports that the XLMS data is not an artifact, and it reflects interactions observed before by us and other laboratories. Our XLMS has the added value of cross-links, indicating potential direct interactions and interacting domains.

To address this in the manuscript, we included: Results, lines 115-122: “Enrichment analysis comparing the immunoprecipitations in cells expressing V5-tagged PIP5Pase or non-tagged PIP5Pase (as a control) confirmed the enrichment of PIP5Pase (\log_2 fold-change = 3.66, p-value = 0.00029) (Supplementary Fig. 2 and Supplementary Table 2). To validate the cross-link dataset, we analyzed whether the cross-linked proteins overlapped with PIP5Pase co-immunoprecipitated proteins that we previously identified without cross-linking¹³. The data show inter-protein cross-links for the top 30 proteins previously identified in PIP5Pase immunoprecipitations without cross-linking. The overlapping dataset confirms PIP5Pase enrichment and suggests potential direct interactions between proteins (Supplementary Fig. 2).”, and from lines 163-167: “Many cross-linked proteins were previously shown to co-interact^{13, 33}, such as PIP5Pase, RAP1, HDAC3, BDF2, ZCWI, EAF6, TRF, DNA PolQ, SMC1, TRF, H2A, NUP158, and GLE2. Their reproducibility here validates the cross-link dataset; however, the cross-link data indicate potential direct partners and their protein domains involved in interactions, thus providing a more refined dataset.”

Lines 168-178: “Immunoprecipitations of V5-tagged PIP5Pase co-immunoprecipitated RAP1 (Supplementary Fig. 2 and Supplementary Table 2), confirming earlier observations^{13, 21, 33}. We obtained 373 peptides for PIP5Pase, of which only six were cross-linked. In contrast, we obtained 241 peptides for RAP1, of which 39 were cross-linked, indicating that

PIP5Pase is a particularly poorly cross-linked protein by DSS. In vitro cross-linking after PIP5Pase immunoprecipitation identified 57 peptides for PIP5Pase but did not result in additional cross-links (Supplementary Table 5). Because we obtained 158,718 total peptides, of which 66,820 were cross-linked, it is unlikely that this reflects inefficient digestion of cross-linked proteins, but perhaps an intrinsic limitation of this protein in being cross-linked by DSS. The limitations could be due to its folding, thereby decreasing the exposure of DSS cross-linkable amino acids. Nevertheless, its enrichment revealed a chain of cross-links, indicating a protein interaction network (Fig. 1F and Fig. 2C).”.

Results, lines 122-125: “*The larger number of proteins identified by XLMS compared to those identified by standard (not cross-linked) immunoprecipitations¹³ may result from cross-links between proximal or transiently interacting proteins not detected by standard approaches.”.*

Discussion, lines 411-415: “*The immunoprecipitation of PIP5Pase (373 peptides) pulled down RAPI (241 peptides); however, we did not detect in vivo cross-links between these proteins, which might be due to the lack of cross-linkable amino acids within the ~20-30Å constraints of DSS. Nevertheless, the two proteins co-immunoprecipitate as reported in this work and previously^{13,21,33}, and we confirmed their direct interactions by SPR.”*

#2) We included **Supplementary Table 4** with the list of 204 cross-linked proteins and added to the spreadsheet the cross-links of the 25 proteins (Multiple XLs >6 proteins, and multiple XLs = 3-6 proteins).

#3) Regarding DSS mass, we clarified the details in Methods, **lines 520-551:** “*the DSS 136.068 mass for two peptide crosslinks, and 156.07864 for mono links”.* And in line 491: “*A total of 15 µg of protein was used for digestion.”*

#4) As for procyclic experiments, we apologize for the confusion; all experiments were performed in bloodstream forms. We removed any description of procyclic forms.

3) Figures and Tables (XL-MS):

Fig. 1E – 918 XL: What does this number represent? Number of cross-links? Proteins?

Author: Number of cross-links. We included it in the Figure 1 legend. **Lines 971-972:** “*Cross-links of a subset of nuclear proteins (377 proteins and 918 cross-links) identified by PIP5Pase immunoprecipitation organized by functional categories.”.*

Fig. S1A: Why does the number of cross-linked proteins increase progressively and then drop sharply at experiment number 12? Please clarify. **Author:** The previous version graph showed only nuclear proteins. We replaced them with graphs that show the reproducibility of the total XLMS dataset (Supplementary Fig. 1A), of the nuclear proteins (Supplementary Fig. 1B), and frequency of total protein detection (Supplementary Fig. 1C). We added the code used for the analysis to GitHub (<https://github.com/cestari-lab/>). The dataset 1-10 was performed by *in vivo* cross-linking of cells followed by PIP5Pase-V5 immunoprecipitation, whereas the dataset 11 was performed by PIP5Pase-V5 immunoprecipitation followed by *in vitro* cross-linking. We increased the stringency of the XLMS data presented from 0.05 FDR (before) to 0.01 FDR (now), hence some differences in the reproducibility index, although it did not affect the identification of those proteins reported in Fig. 1 and Fig. 2.

3. Hi-C and Pore-C Data:

Table S4: Does it represent the sum of replicates? This should be clarified in the caption. **Author:** Yes. We have clarified that this is the sum of replicates in the Table legend, which is now Supplementary Table 7 (see Supplementary Information file).

Fig. 3C: How were the A and B compartments defined? The authors mention using FANC software, but no details are provided regarding the criteria or thresholds. This must be described in the Methods section.

Author: We included in the **Methods, lines 619-624:** “*Compartments were calculated using FAN-C⁵⁶. Briefly, we used the fanc compartments command to identify A and B compartments from a 50 kb resolution Hi-C matrix, where A has positive values and B has negative values. Compartments were uniformized based on the genome's GC content (option -g), with A having high GC content and B having low GC content. The eigenvector values were then written to the file (option -v). Then, using RNA-seq of *T. brucei* bloodstream forms¹⁴, we associated transcribed and silent regions according to A and B compartments, respectively.”.*

Fig. 3D: How were short-range and long-range contacts defined? Please specify the distance cutoffs used.

Author: Short-range contacts were defined as those lower than the cutoff, and long-range contacts as those above the cutoff. The cutoff, maximum distance between two intervals in the chromosome, was 100,000 Kb. We use the recommended values suggested by Magnitov et al. 2022, BMC Bioinformatics. We included this information in the

Methods, lines 639-641: “To quantify interactions among compartments, we performed analyses with Pentad⁵⁸. We used the `get_pentad_cis.py` script with a KR-balanced matrix and a cutoff distance of 100,000 kb, i.e., short-range contacts are lower than the cutoff and long-range contacts are higher than the cutoff.”

Which version of the *T. brucei* genome (FASTA file) was used? **Author:** The data were mapped to the *T. brucei* 2018 genome (Muller et al. 2018, Nature, Ref #7 in the manuscript). There are two haplotypes available, which differ in the subtelomeric regions. We analyzed against both haplotypes, but show results for forkA for simplicity. We added to **Methods, line 600:** “*T. brucei* 427 genome⁷ (data shown for haplotype A)...”

What Illumina sequencer and read length were used for Hi-C? **Author:** We included in **Methods, lines 572-573:** “Libraries were sequenced at GenomeQuebec using an Illumina NovaSeq sequencer for paired-end 150 bp sequencing.”

Regarding Pore-C, there is no description of the protocol in the Methods section. Additionally, the text states that “DNA fragments ranging from 100–450 bp were size-selected”, which seems inconsistent with the expected advantages of Pore-C, typically designed to capture much longer DNA fragments. At line 186, the authors report an N50 of 1.54 kb — please clarify how this was achieved and discuss whether such short fragments truly justify the use of Pore-C over conventional Hi-C. **Author:** We included a new session in the Methods with a complete description of Pore-C, see **lines 575-600**, headed “*Pore-C library and sequencing*”. We also included a description of the computational analysis, **lines 602-651**, headed “*Hi-C and Pore-C computational analysis*.” The fragmentation of 100-450 bp was done for Illumina only. No fragmentation was done for Pore-C.

4. Concerns About Definitions and Interpretation:

The distinction between “compartment” referring to A/B compartments (Hi-C based) and core/subtelomeric regions (genomic context) is unclear throughout the manuscript. For example, in Fig. 3G, the authors refer to two compartments associated with transcribed and silent regions. Are these the same as the A/B compartments determined by FAN-C? Has it already been demonstrated that A and B compartments correspond to core and subtelomeric regions in *T. brucei*? If so, please cite the relevant studies. If not, the authors should explicitly show whether or not such correspondence exists. How were sub-TADs defined? Also long and short interactions? What criteria were used?

Author: The presence of chromosome compartments was first observed by Muller et al. 2018, Nature. This reference was included in the **Introduction, lines 66-67:** “Chromatin conformation capture experiments showed that core and subtelomeric regions of *T. brucei* chromosomes form distinct compartments⁷...”. We expanded the analysis using FAN-C to identify A and B compartments and organize them based on GC-content and RNA-seq, formally showing their association with transcribed core and silent subtelomeric regions. We clarified this in the **Results, lines 200-209:** “Using a correlation matrix analysis³⁵, we found that each chromosome contained about two or three compartments with an average length of 1.2 Mb (± 1 Mb, range of ~1-4 Mb), with core transcribed regions forming distinct compartments from silent subtelomeric regions (Fig. 3C and Supplementary Fig. 3), analogous to A and B compartments, respectively¹. Using GC-content, we oriented the core and subtelomeric compartments, as subtelomeric regions have lower GC content (~40%) than core regions (~50%). Moreover, using RNA-seq from *T. brucei* bloodstream forms of the SM427 strain¹⁴, we assigned transcribed and silent regions to A and B compartments, respectively (Fig. 3C and Supplementary Fig. 3). Notably, in *T. brucei*, the organization of A and B compartments correlates well with core and subtelomeric regions for Mb-long chromosomes.”. We included A and B terminology associated with core and subtelomeric regions throughout the text and figures.

We removed mention of sub-TADs, as they are smaller TADs within larger TADs, and there are no specific criteria for the terminology use based on size. TADs vary between ~150Kb to 1Mb in various organisms (Dixon et al. 2012, Nature; Beagan and Phillips-Cremens, 2020, Nat Genetics; Szabo et al. 2019, Science Advances; Eser et al. 2017, PNAS; Torosin et al. 2022, Mol Biol Evol).

For short and long interactions, we added to **Methods, line 640** (as indicated above): “a cutoff distance of 100,000 kb, i.e., short-range contacts are lower than the cutoff and long-range contacts are higher than the cutoff”. We also added this information to the Figure legends 3 and 5.

Fig. 4G appears to be missing. **Author:** We corrected it. It should have been Fig. 5G.

Fig. 4E is referenced before Fig. 4A-C; please check the figure order. Also, confirm whether the color descriptions (red vs. green) are correct. **Author:** We changed the order and corrected the legend. The red corresponds to boundaries, whereas green corresponds to transcriptional hubs.

In Fig. 3G (inset): It is well established that many chromatin-associated proteins localize to divergent strand switch regions (dSSRs) in *T. brucei*. The authors should discuss whether the observed enrichment at boundaries reflects an

association with dSSRs. It would be informative to annotate PTU boundaries directly on the heatmaps shown (e.g., Fig. 3G). Interestingly, proteins like BDF2, HDAC, HAT1, and ZCW1 flank RAP1 at boundaries. This is a noteworthy observation that deserves further discussion. Does this pattern occur at multiple boundaries? Are these proteins consistently located at TAD boundaries? **Author:** We included in Fig. 3G the PTUs (black for genes in forward and grey reverse orientation), and ChIP-seq of BDF2 to inform the reader of transcription start regions (TSRs) since BDF2 binding sites were reported in these regions (Staneva et al. 2021, Genome Research). The graph in Fig. 3H shows the average data of all chromosome compartment boundaries. The proteins are present in most compartment boundaries, either flanking or near RAP1. We included Supplementary Fig. 5 with ChIP-seq data for all proteins on all Mb-long chromosomes.

We included in the **Results, lines 246-248:** “*The presence of some of these proteins at the compartment boundaries could be due to some overlap with TSRs, but they might also have a role in chromatin modification at these regions.*”. And to the **Discussion, lines 377-384:** “*We found specific chromatin-binding proteins at the boundaries between core (A) and subtelomeric (B) compartments, such as RAP1, BDF2, HAT1, HDAC1, SCC1, and ZCW1. Not all boundaries contain all proteins, but a combination of them, with RAP1 present in nearly all of them. If a boundary is at the centromere, then KKT2, RAP1, and SCC1 are present. Some of these factors (BDF2, HAT1, HDAC1, ZCW1) are also present at TSRs³³, some of which are within the boundaries. These proteins are typically neighbouring RAP1. The presence of these factors at the compartment boundaries could coincide with the TSRs, or perhaps they have a direct role in chromosome compartment boundaries associated with their function in chromatin acetylation or methylation.*”

Regarding TAD boundaries, TAD boundaries typically encompass 1-4 PTUs, and in some cases overlap with these proteins. We included in the **Discussion, lines 348-354:** “*A possible role for TADs and loops in T. brucei might be to organize subsets of PTUs around transcriptional hubs. This arrangement could provide efficient transcription and RNA processing by grouping and recycling the transcription machinery around contact regions. Previous work also suggested TSRs may form distinct inter-chromosomal transcription hubs, which may facilitate RNA polymerase II transcription²⁴. Proteins present at TSRs, such as BDF2, HAT1, and HDAC1, may help organize chromatin contact in these regions and facilitate recruitment of the transcription machinery.*”

5. Repetitive Regions and Potential Artifacts:

The authors should investigate whether boundaries are enriched in repetitive elements. Could some of the observed boundary-to-boundary interactions be artifacts resulting from the presence of highly repetitive subtelomeric regions? It would strengthen the manuscript if the authors analyzed sequence similarity between boundaries, searched for motifs, and discussed how they handled multimapping reads during Hi-C and Pore-C data processing. **Author:** For both Hi-C and Pore-C, we used pairtools parse (Hi-C) or parse2 (Pore-C), which only considers uniquely mapped reads to identify contacts. We included this in the **Methods, lines 605 and 609:** “*and filtering uniquely mapped reads and removal of duplicates.*”. Analysis of boundary sequences revealed similar types of sequence motifs, which are, in general (TTA)_n(CCA)_n, with some variations. We included Fig. 4F summarizing the analysis and Supplementary Fig. 6 (pdf file) with detailed analysis of the sequence repeats. The repeats are similar to those we reported for RAP1 binding sites (Touray et al. 2023 eLife) in 70 bp repeats. Hence, to ensure RAP1 binding to these regions was not an artifact, we performed ChIP-seq analysis at various conditions. Note ChIP-seq was done with nanopore sequencing rather than Illumina for better mapping to these regions. We analyzed RAP1 ChIP-seq all reads (primary+secondary), only primary reads, and reads mapping with mapQ = 10, which is very stringent. There were no significant changes in the mapping, indicating that RAP1 binding to boundaries is not an artifact of read alignment (Supplementary Fig. 7). Although repeated motifs are found, the sequences are divergent sufficiently to result in unique alignments. We included in **Results, lines 262-268:** “*Analysis of boundary sequences revealed an enrichment in (TTA)_n(CAC)_n repeats or sequences with stretches of C/G repeats (Fig. 4F and Supplementary Fig. 6). These sequences resemble those found in the 70 bp repeats within the ESs, but also the AT-rich centromeric regions or telomeric repeats, all of which are binding sites for RAP1¹⁴. Hi-C and Pore-C reads included only uniquely mapped reads. Analysis of RAP1 ChIP-seq (performed with Oxford nanopore sequencing) with stringent mapping quality (mapQ = 10) retained RAP1 binding to boundaries and subtelomeric regions (Supplementary Fig. 7), indicating that its DNA binding is not an artifact of alignment.*”

6. Data Normalization:

Tables S3 and S4: The number of Hi-C contacts varies significantly between replicates (e.g., PIP5ase Tet+ ranges from ~70K to ~116K contacts), seemingly independent of the number of reads. Is this expected? Please explain. Related to this, before calculating log₂FC values (Fig. 5B), shouldn't the authors normalize for the same number of reads between Tet+ and Tet- samples? There are several established metrics for comparing Hi-C matrices that account for

varying sequencing depths — please clarify which normalization strategy was used. **Author:** We obtained about ~300M to 600M total reads for each Hi-C biological replicate, and contacts varied between 66M and 116M. The variation reflects differences in libraries and alignments because of variations in biological replicates. The experiments were performed with *T. brucei* cells growing on different days, and libraries were also prepared on different days, as we wanted to account for biological variations. Hence, variation reflects true biological differences among replicates; nevertheless, the number of contacts was high and sufficient for analysis. We included some analysis in Supplementary Fig. 8. The data show principal component analysis of WT (SM527) and conditional null PIP5Pase Tet⁺ and Tet⁻, demonstrating some divergence between Tet⁻ compared to SM427 or Tet⁺. The counts vs distance plots show a typical distribution of contacts among the datasets, indicating no abnormalities in the data. Moreover, we included plots with examples of chromosome compartments in SM427 and Tet⁺, which are comparable, but both distinct from Tet⁻. The same can be observed for TADs (see Fig. 5F), indicating that SM427 and Tet⁺ data are comparable, and differences are observed with Tet⁻.

As for normalizations, libraries were normalized for the smallest dataset among the groups (SM427, Tet⁺, and Tet⁻). We included this in the **Methods, lines 611-619**: “*Matrices were balanced using the Knight-Ruiz (KR) method, as implemented in hicCorrectMatrix. For comparison, we also corrected matrices using iterative correction and eigenvector decomposition (ICE) with hicCorrectMatrix, using filterThreshold values of -1.5 and 5. We did not observe significant differences between KR and ICE balancing upon matrix visual and content inspection. Hence, further analysis was obtained using the KR balancing method. To compare different datasets, matrices were normalized to the condition with the smallest matrix using the hicNormalize tool. To compare, we also normalized matrices from 0 to 1, and did not detect differences between the two normalization results. The results shown were obtained by normalizing to the smallest matrix.*”

Reviewer #2 (Remarks to the Author):

This study provides significant contributions to the scientific understanding of chromosome organization and gene silencing in *Trypanosoma brucei*. It reveals that subtelomeric gene silencing—a mechanism essential for antigenic variation and immune evasion—relies on the spatial segregation of chromatin into distinct chromosome compartments. By identifying and characterizing key chromatin-associated proteins such as RAP1, the work demonstrates that these components regulate the organization of repressed and active regions of the genome. Moreover, it establishes a novel connection between phosphoinositide signaling and chromatin structure, suggesting that lipid-mediated regulation may play a broader role in nuclear architecture. These findings provide new insights into gene regulation mechanisms in eukaryotic parasites and potentially other organisms, representing a significant step toward understanding epigenetic control in complex pathogens. Although this work advances our knowledge of chromatin compartmentalization, we have questions and comments we detail below: **Author:** We appreciate the reviewer's valuable time and evaluation of our work.

- The authors must specify how the Hi-C data matrices were normalized. The algorithm/method used in the Methods section must be detailed, as normalization is a critical step in the analysis that significantly impacts the results. **Author:** Matrices were balanced using Knight-Ruiz method implemented in HicExplorer, and groups (WT, Tet⁺, and Tet⁻) were normalized for the smallest library. We compared normalizing the groups from 0-1 and found no differences in the results. We included the information in the **Methods, lines 611-619**: “*Matrices were balanced using the Knight-Ruiz (KR) method, as implemented in hicCorrectMatrix. For comparison, we also corrected matrices using iterative correction and eigenvector decomposition (ICE) with hicCorrectMatrix, using filterThreshold values of -1.5 and 5. We did not observe significant differences between KR and ICE balancing upon matrix visual and content inspection. Hence, further analysis was obtained using the KR balancing method. To compare different datasets, matrices were normalized to the condition with the smallest matrix using the hicNormalize tool. To compare, we also normalized matrices from 0 to 1, and did not detect differences between the two normalization results. The results shown were obtained by normalizing to the smallest matrix.*”

- The heatmaps presented contain several regions with limited contact information. Which resolution was used to analyze domain formation and loop structures? The authors should indicate the resolution of each heatmap, either within the figures or in the corresponding legends, as well as specify the resolution used for each type of analysis. **Author:** The figures show heatmaps at 10 Kb and 50 Kb resolutions. We included this information in all Figure legends. The resolution to analyze loops was 10 kb, as recommended (Ramirez *et al.*, 2018, *Nature Communications*). We included the information in the **Methods, lines 635-637**: “*For loops, we used hicDetectLoops, using a 10 Kb matrix, maximum loop distance of 2 Mb, p-value of 0.025, peak region width of 2 bins, and a p-value pre-selection threshold of 0.1.*”. For TADs, we performed analysis with matrix resolutions at 1Kb, 10 Kb, and 50 Kb. We have included this in the **Methods, lines**

625-634: “TADs were searched using *hicFindTADs* using a 1Kb, 10 Kb and 50 Kb resolution matrix, a delta of 0.01, threshold of 0.05, multiple test correction set to false discovery rate, minimum and maximum depth of 3 times matrix resolution and 10 times resolution, respectively, and steps equal to matrix resolution, using a KR balanced matrix. Afterwards, identified TADs were merged using *hicMergeDomains*. We also analyzed TADs and loops using an ICE balanced matrix, with the same parameters as above, without significant differences between ICE and KR, although KR balancing resulted in more reliable detection of TADs after their inspection and comparison among datasets. The results shown in the manuscript and all subsequent analyses were performed using a KR balanced matrix. To compare inter- and intra-TADs, we used the *hicInterIntraTAD* tool with the TAD domains identified by *hicFindTADs* (as specified above).”.

- Are the contact frequencies in figures 3B, 3C, 3G, 5A, 5B, 5C, 5G, and 6A normalized? If so, please indicate them in the figures. **Author:** Matrices were balanced using KR methods, and comparisons between groups Tet + vs Tet- were performed with normalization to the smallest matrix. We included it in the Methods, as indicated in the answer to the question above. We included this information in the Fig. legends.

-It would be interesting to include additional examples similar to Figure 3C to illustrate how compartments are organized across other chromosomes. The authors are encouraged to provide more examples as supplementary figures. **Author:** We included Supplementary Fig. 3 with plots showing compartments for all chromosomes with some ChIP-seq data included. Supplementary Fig. 5 shows Hi-C and ChIP-seq data for all proteins analyzed. Supplementary Fig. 9 has additional examples comparing compartment disruption, and Supplementary Fig. 10 shows Hi-C after PIP5Pase knockdown with ChIP-seq for RAP1 and RNA-seq comparing PIP5Pase WT vs Mutant to indicate subtelomeric compartment transcription.

- The authors should include a supplementary table listing the predicted domains and loops to ensure this information is accessible to the scientific community. **Author:** We included Supplementary Table 8 (Excel file) with all identified loops and TADs. The Table includes TADs identified with Hi-C and Pore-C data for 1Kb, 10K, and 50 Kb resolution, as well as data including merged TADs from different resolutions and their relatedness.

-The authors should present additional genomic regions to better assess the distribution of chromatin-associated proteins, rather than limiting the analysis to a single locus as shown for BDF2, HAT1, HDAC1, and ZCW1. Furthermore, it would be helpful to visualize RAP1 distribution across other chromosomes. Please include supplementary figures with additional examples to support a more comprehensive evaluation. **Author:** Figure 3H is a summary of all chromosomes. We included Supplementary Fig. 5 to show data for all chromosomes.

-Figures 3e and 5e require a more detailed explanation. The authors should clarify how these figures were generated, including the methods used for their calculation, and clearly describe what the data represent. **Author:** These analyses were performed with *Pentad* (Magnitov et al. 2022, *BMC Bioinformatics*). We also included the citation in the References section, which contains a detailed explanation of the method and its application. We included a detailed explanation of the analysis in the **Methods, lines 639-644:** “To quantify interactions among compartments, we performed analyses with *Pentad*⁶⁶. We used the *get_pentad_cis.py* script with a KR-balanced matrix and a cutoff distance of 100,000 kb, i.e., short-range contacts are lower than the cutoff and long-range contacts are higher than the cutoff. The A and B compartments, as explained above, were used to define compartments. For trans-contacts, we used *get_pentad_trans.py*. Compartment strengths for cis and trans-contacts were analyzed using *quant_strength_cis.py* or *quant_strength_trans.py*, respectively.”. In the Results, we expanded on this explanation, **lines 209-211:** “We then quantified contacts within compartments (intra-chromosomal contacts) and between compartments of different chromosomes, i.e., core vs core (A vs A), core vs subtelomeric (A vs B), and subtelomeric vs subtelomeric (B vs B). We found... ”.

-Line 240, correct 4G for 5G. **Author:** Done

- Although the authors have made their scripts available on GitHub, they should also specify the algorithms used and provide a general description of the RNA-seq and ChIP-seq analysis methods within the manuscript. **Author:** Included in the **Methods, line 653**. Session headed as “ChIP-seq and RNA-seq computational analysis”,

- The RNA-seq results should be better presented, as the current figures do not allow for clear visualization of the +/- Tet effect on global gene expression within the compartments, as described in the text. **Author:** We replaced the RNA-seq in

Fig. 6A to make it clearer (green shows upregulation and red downregulation). Also, we included the data in Supplementary Fig. 3 showing the effect for each compartment on all Mb-long chromosomes. Supplementary Fig. 10 shows figures like Fig. 6A but for other chromosomes.

- The formation of chromatin domains in both *T. cruzi* and *T. brucei* has been previously described (DOI: 10.1038/s41564-023-01483-y), including a comparative analysis of domain organization across different life stages of *T. brucei*. The authors should incorporate this background into the Introduction or Discussion to provide better context for the current findings. **Author:** We included in the **Discussion, lines 358-359**: “...*Although analysis of the same dataset by others reported the presence of such domains, also found in T. cruzi*⁴⁵.”

- The Hi-C data is not available at the accession code provided in the manuscript. **Author:** Hi-C sequencing data were deposited in the SRA with BioProject identification PRJNA1198910, as indicated in Resource availability. The reviewer link is here: <https://dataview.ncbi.nlm.nih.gov/object/PRJNA1198910?reviewer=bbjk2g4po2njrbr517ca2po4g>
Scripts to reproduce the data are here: <https://github.com/cestari-lab/>

- It is important for the scientific community to establish a unified nomenclature for chromatin domains in trypanosomes. *T. cruzi* and *T. brucei* previously described these domains as Chromatin Folding Domains (CFDs). The authors should clarify their choice to refer to them as TADs and discuss how this terminology relates to or differs from previously proposed definitions. **Author:** The terminology TAD has been used in all fields of biology (human, yeast, *C. elegans*, *Drosophila*, etc) to describe a segment of genomic DNA sequence co-interacting more frequently than with sequences outside the specific segment as obtained from Hi-C experiments (Szabo et al. 2019, Science Advances). The computational algorithms used in this work were designed to identify TADs using Hi-C data, and thus, we report them as found. The term CFDs has only been used in a single paper (Diaz-Viraqué et al. 2023, Nat Microb). The authors used algorithms designed to identify TADs (i.e., TADtools) in Hi-C data, but renamed the identified TADs to CFDs based on their length (~129 Kb). The term CFD refers to TADs identified with algorithms and methodology established for TADs in the broader field of biology (Dixon et al. 2012, Nature; Beagan and Phillips-Cremens, 2020, Nat Genetics; Szabo et al. 2019, Science Advances). For example, TADs of ~200 Kb were identified in yeast (Eser et al. 2017, PNAS), and TADs of 135 to 215 Kb in *Drosophila* species (Torosin et al. 2022, Mol Biol Evol), which are similar in size to those identified in *T. cruzi*. Díaz-Viraqué et al. paper (Fig. 5B) shows *T. brucei* TADs ranging 150-400 Kb, which is consistent with what we report and found by others (Dixon et al. 2012, Nature; Rowley and Corces, 2018, Nat Rev Genet). Hence, TADs seem to have a typical range of 100-1000 Kb in various organisms. We opted for the TAD terminology because it is consistent with the methods (Hi-C data and algorithms used) and aligns with the term usage in the broader field.

- Throughout the manuscript, the authors refer to PTUs (polycistronic transcription units), when in our view they should instead refer to DGC (directional gene clusters). Although the two concepts are related, they are not interchangeable: PTUs are defined by their transcriptional activity, whereas DGCs refer to blocks of coding genes oriented in the same direction, regardless of whether they are actively transcribed. **Author:** As suggested, we included directional gene clusters when referring to non-transcribed regions, see **Introduction, line 40-41**: “*subtelomeric regions contain largely silent variant surface glycoprotein (VSGs) genes arranged as directional gene clusters...*”. The term PTUs applies to core regions that are transcribed in *T. brucei*, and for these cases, the PTU terminology is accurate and consistent with its use in the field (Muller et al. 2018, Nature; Staneva et al. 2021, Genome Res; Rabuffo et al. 2024, Nat Commun; Siegel et al. 2009, Genes and Development; Reynolds et al. 2016, PLOS Genetics; Jensen et al., 2009, BMC Genomics; Maree et al. 2017, Epigenetics and Chromatin; Novotna et al. 2025, Nature Communications; Schulz et al. 2016, PLOS Genetics).

Reviewer #3 (Remarks to the Author):

I co-reviewed this manuscript with one of the reviewers who provided the listed reports. This is part of the Nature Communications initiative to facilitate training in peer review and to provide appropriate recognition for Early Career Researchers who co-review manuscripts. **Author:** We appreciate your participation, time, and input.

Reviewer comments in black, author answers in blue.

Reviewer #1 (Remarks to the Author):

The authors have satisfactorily addressed most of the questions. However, as I mentioned before, although the manuscript indeed contains new data that advances the field, it is interspersed with results already shown either by the authors themselves or by others. Additionally, the compartments findings has a critical semantic issue that must be addressed to avoid confusion. Thus, it is up to the editor to evaluate whether this constitutes a justified novelty to be published in this journal.

The major semantic issue of this manuscript concerns the previous definition of compartments (subtelomeric/telomeric vs. core compartments), and now the classical definition of A and B compartments from the Hi-C topological organization; and the fact that both compartment definition is indeed referring to the same class of genes/location in the genome. As it is currently written, parts of the text may give the impression that the authors are simply reiterating what has already been shown by Nicolai Siegel's group (Müller et al.; Rabuffo et al., etc.). However, I understand that the authors are actually referring to the A and B compartments identified using the standard Hi-C bioinformatics pipeline. To avoid confusion, I strongly recommend that this compartment definition distinction be clearly stated at the beginning of the manuscript.

For example, in Muller et al. (paper from Nicolai Siegel's lab) they showed that "Genome-wide chromosome conformation capture (Hi-C) reveals a distinct partitioning of the genome, with antigen-encoding subtelomeric regions that are folded into distinct, highly compact compartments." Thus, the existence of core compartments (highly transcribed, fewer DNA-DNA contacts) and antigen-encoding subtelomeric compartments (silent, more DNA-DNA contacts) was already established. I understand that the main advance here is that their Hi-C and Pore-C assays now reveal the presence of A and B compartments, TADs and loops (although Viraque et al. 2024 also showed part of this) by applying standard Hi-C computational pipelines. Therefore, statements such as "- We show that *T. brucei* chromosomes are organized into core transcribed and silent subtelomeric compartments,..." (abstract - line 17), or at lines 198–199 "Thus, we performed Hi-C and ChIP-seq analysis in *T. brucei* bloodstream forms. *T. brucei* chromosomes are organized into core transcribed and subtelomeric repressed regions, with some chromosomes containing ESs (Fig. 3A)." do not seem appropriate without citing the original work. In my view, the novelty is that they confirm the presence of compartments already reported and now refine this observation using proper bioinformatic analysis of Hi-C data (such as FANC). **Author:** We included the citation as indicated (ref #7, line 199). As for the Abstract, we changed to: "*We show that T. brucei core and subtelomeric chromosome compartments are separated by distinct boundaries ...*".

A definition of A and B compartments is provided in the Introduction, lines 33-34: "*Eukaryote chromosomes are typically organized into megabase (Mb) scale compartments defined as transcribed A compartment (euchromatin) and silent B compartment (heterochromatin)*^{1, 2}". Also, in the third paragraph (lines 66-68), we address it *T. brucei*: "*Chromatin conformation capture experiments showed that core and subtelomeric regions of T. brucei chromosomes form distinct compartments*⁷, which correlate with their transcribed and repressed state, respectively."

Another example: At line 208 they write: "Notably, in *T. brucei*, the organization of A and B compartments correlates well with core and subtelomeric regions for Mb-long chromosomes." A more precise statement would be something like: "Notably, in *T. brucei*, the organization of A and B compartments correlates well with core and subtelomeric regions for Mb-long chromosomes – 3D compartments already described before (Muller et al.)." **Author:** We changed as suggested; lines 207-209: "*Notably, in T. brucei, the organization of A and B compartments correlates well with previously described core and subtelomeric compartments*⁷ for Mb-long chromosomes, with B compartments also including silent ESs".

Another important point:

In their reply, the authors state: "The binding of RAP1 became evident to us only when analyzing RAP1 ChIP-seq using a genome version in which subtelomeric regions were co-assembled with core regions." However, it has

already been shown that RAP1 binds (and unbinds) to subtelomeric regions in TbPIP5Pase mutants (see Fig. 5 and its legend of Touray et al. 2023). **Author:** The published data (Touray et al. 2023, *eLife*, Fig. 5) refer to ~10-80 Kb-long **telomeric expression sites (ES)**, each of which contains a single VSG gene and can be transcribed. It shows that RAP1 binds to ESs at the 70 bp repeats and telomeres. Please note that they are not the megabase-long subtelomeric compartments housing hundreds of silent VSG genes reported here.

Finally, the information that perturbation of inositol metabolism affects the expression of silent VSGs was already published by this group: “Conditional knockdown of phosphatidylinositol 5-kinase (TbPIP5K) or phosphatidylinositol 5-phosphatase (TbPIP5Pase) or overexpression of phospholipase C (TbPLC) derepresses numerous silent ESs in *T. brucei* bloodstream forms.” (doi: 10.1073/pnas.1501206112). The way it is written, makes it sound like a novel finding (see abstract, lines 22–23: “Inactivation of the RAP1 regulator, phosphatidylinositol phosphate 5-phosphatase, removes RAP1 from boundaries and subtelomeric compartments, disrupting chromatin compartment contacts and activating all VSG genes.”). The truly new finding here is that they show that inactivation of phosphatidylinositol phosphate 5-phosphatase removes RAP1 from boundaries and subtelomeric compartments, disrupting chromatin compartment contacts. The activation of VSG expression, however, had already been reported by the group. **Author:** To clarify, we describe in detail in the Introduction, lines 58-65, how PIP5Pase regulates RAP1 silencing of ESs; hence, we are not claiming this here. The text in the Abstract contextualizes it for the reader to understand the findings. The findings reported here support a mechanism regulating silencing of megabase-long VSG-rich subtelomeric compartments, which is significant and a new conceptual advance.

Minor points:

What does the circle composed of several small green circles in Fig. S2C represent? **Author:** Hypothetical proteins. It follows the colour-coded legend. We included an arrow to facilitate identification.

In the virtual 4C analysis of chromosomes 11 and 10 (Fig. 4C and 4), the figure could be improved, perhaps by including a scheme showing where the chromosome 10 and 11 boundaries lie on the x-axis. **Author:** Included. The suggestion is much appreciated.

In Fig. 5C, is there a normalization issue? The Tet– Hi-C matrix clearly shows no compartments, yet the plots suggest boundaries. Note also that the scales of Tet+ and Tet– are not the same, which makes the comparison even more difficult. **Author:** There is no normalization issue. Compartment contacts were disrupted by PIP5Pase knockdown; however, boundaries are still preserved. We emphasized this in the Results, line 289: “Analysis of chromosome compartments showed significant disruption of contacts within the compartments (Fig. 5C-D and Supplementary Fig. 9), although boundaries were preserved.”. We adjusted line graphs to have the same scale for Tet + and Tet –.

What is the phenotype of Tet– cells? (viability? infectivity?). **Author:** PIP5Pase does not affect cell viability within 24h, as demonstrated by cell growth and live/dead assays; growth effects begin within 48h-72h (Cestari and Stuart, 2015, PNAS). All experiments, Hi-C, ChIP-seq, and RNA-seq, were performed at 24h. We included, in line 280: “PIP5Pase does not affect cell viability within 24h²¹; hence, ...”.

PIP5Pase knockdown impairs infection of mice (Cestari et al. 2016, *Cell Chem Biol*). We included in the Discussion, lines 442-443: “PIP5Pase is essential for *T. brucei* infection¹⁵, and considering its regulatory roles, it could be explored as a drug target.”

Lines 301–302: “The data indicate that PIP5Pase also coordinates subtelomeric compartment interactions, which may occur via the association...” → this should read “the data SUGGEST that PIP5Pase...” since the authors have not shown by orthogonal methods that proteins of the PIP5Pase network (except for BDF2, HAT, etc., which also bind elsewhere in the genome) indeed bind to compartment boundaries (e.g., by ChIP-seq or KO). **Author:** Edited as recommended, replaced *indicate* to *suggest* in line 303.

Lines 321–323 seem contradictory:

“The data suggest that contact among boundaries is not necessary to repress silent subtelomeric regions, but it may

help demarcate core and subtelomeric compartments. The data indicate that the assembly of chromosome compartments is essential for silencing subtelomeric VSG genes.” **Author:** We removed the second sentence.

Fig 6^a : RNA-seq ratio between wt/mutant seem opposite. **Author:** This is discussed in the manuscript; lines 319-321: “There was a slight decrease in the expression of core genes (Fig. 6A, F, Supplementary Fig. 3 and 10), which might reflect the disruption of core compartment structures, such as TADs and loops, perhaps affecting the efficiency of core gene transcription or RNA processing.”.

Reviewer #2 (Remarks to the Author):

The authors have sufficiently addressed the concerns raised in the previous round of review and have incorporated the missing information into the manuscript. However, there remain a few additional comments that we would like to put forward for consideration.

Authors:

"It is important for the scientific community to establish a unified nomenclature for chromatin domains in trypanosomes. *T. cruzi* and *T. brucei* previously described these domains as Chromatin Folding Domains (CFDs). The authors should clarify their choice to refer to them as TADs and discuss how this terminology relates to or differs from previously proposed definitions. **Author:** The terminology TAD has been used in all fields of biology (human, yeast, *C. elegans*, *Drosophila*, etc) to describe a segment of genomic DNA sequence co-interacting more frequently than with sequences outside the specific segment as obtained from Hi-C experiments (Szabo et al. 2019, Science Advances). The computational algorithms used in this work were designed to identify TADs using Hi-C data, and thus, we report them as found. The term CFDs has only been used in a single paper (Díaz-Viraqué et al. 2023, Nat Microb). The authors used algorithms designed to identify TADs (i.e., TADtools) in Hi-C data, but renamed the identified TADs to CFDs based on their length (~129 Kb). The term CFD refers to TADs identified with algorithms and methodology established for TADs in the broader field of biology (Dixon et al. 2012, Nature; Beagan and Phillips-Cremens, 2020, Nat Genetics; Szabo et al. 2019, Science Advances). For example, TADs of ~200 Kb were identified in yeast (Eser et al. 2017, PNAS), and TADs of 135 to 215 Kb in *Drosophila* species (Torosin et al. 2022, Mol Biol Evol), which are similar in size to those identified in *T. cruzi*. Díaz-Viraqué et al. paper (Fig. 5B) shows *T. brucei* TADs ranging 150-400 Kb, which is consistent with what we report and found by others (Dixon et al. 2012, Nature; Rowley and Corces, 2018, Nat Rev Genet). Hence, TADs seem to have a typical range of 100-1000 Kb in various organisms. We opted for the TAD terminology because it is consistent with the methods (Hi-C data and algorithms used) and aligns with the term usage in the broader field."

ANSWER:

The use of algorithms designed for TAD identification does not in itself justify calling the domains observed in trypanosomes TADs. The conceptualization of chromatin structures in microorganisms has evolved differently. Due to differences in genome size and biological features (such as the absence of one promoter per gene and enhancer–promoter regulation), the structural domains identified in trypanosomes were referred to as chromatin folding domains (CFDs). The genomic architecture of these parasites differs substantially from that of multicellular eukaryotes, lacking promoter–enhancer regulation, introns, and large intergenic regions. Accordingly, the domains identified by Hi-C in *T. cruzi* have been described as chromatin folding domains (CFDs), ranging from 20 to 940 kb (average ~129 kb), a size comparable to chromatin interaction domains (CIDs) in bacteria (30–420 kb, average ~170 kb; Dekker et al., 2015; Le et al., 2013). These values differ from those reported for TADs in multicellular eukaryotes (average 880 kb; Dixon et al., 2012), most likely reflecting the reduced extent of non-coding regions in trypanosomatids (including the absence of introns and RNA polymerase II regulatory elements, as well as shorter intergenic and inter-DGC regions). The manuscript would be strengthened if the authors explicitly clarified how their use of the term “TAD” relates to the previously

AUTHORS:

“The failure of other studies in identifying TADs in *T. brucei* may reflect a combination of Hi-C sequencing depth and, thus, the obtained number of contacts and the approach used. Although analysis of the same dataset by others

reported the presence of such domains, also found in *T. cruzi* .”

ANSWER

The original sentence is unclear and needs revision for clarity. The use of “Although” makes the statement vague, especially considering previous comments on chromatin folding domains (CFDs). Reference 45 not only shows CFDs in *Cruzi* but also indicates the presence of CFDs in *T. brucei*, with sizes and magnitudes similar to those reported in *T. cruzi*. Additionally, that work demonstrates conservation of CFD in procyclic and bloodstream forms and a correlation with global transcription, suggesting a common feature of trypanosomatids. Like in *T. cruzi*, there is no exact correlation between DGC and PTU. Therefore, there was no actual ‘failure’ to identify these domains in *T. brucei*. This point should be included in the discussion rather than presenting the results as a new finding. **Author:** We re-phrased the statement and clarified the terminology, as requested, lines 357-360: “*Muller and colleagues did not report TADs in T. brucei*⁷. However, *Diaz-Viraque and colleagues, analyzing the same dataset, found them in T. brucei, but also in T. cruzi*⁴⁵, describing them as chromatin folding domains (CFD). We use the term TAD instead of CFD in strict accord with the methodology and algorithms used, as well as its broad use in other eukaryotes.”

Reviewer #2 (Remarks on code availability):

We have already done it in the previous round.

Reviewer #3 (Remarks to the Author):
